# BARK: A Fully Bayesian Tree Kernel for Black-box Optimization

Toby Boyne [1]   Jose Pablo Folch [1]   Robert Matthew Lee [2]   Behrang Shafei [2]   Ruth Misener [1]

## Abstract

We perform Bayesian optimization using a Gaussian process perspective on Bayesian Additive Regression Trees (BART). Our BART Kernel (BARK) uses *tree agreement* to define a posterior over piecewise-constant functions, and we explore the space of tree kernels using a Markov chain Monte Carlo approach. Where BART only samples functions, the resulting BARK model obtains samples of Gaussian processes defining distributions over functions, which allow us to build acquisition functions for Bayesian optimization. Our tree-based approach enables global optimization over the surrogate, even for mixed-feature spaces. Moreover, where many previous tree-based kernels provide uncertainty quantification over function values, our sampling scheme captures uncertainty over the tree structure itself. Our experiments show the strong performance of BARK on both synthetic and applied benchmarks, due to the combination of our fully Bayesian surrogate and the optimization procedure.

## 1. Introduction

Bayesian optimization (BO), which finds the optimal conditions of expensive-to-evaluate black-box functions (Frazier, 2018; Shahriari et al., 2015; Wang et al., 2023c), is a leading approach for data-driven design of experiments. BO applications range from molecule design (Griffiths et al., 2024) to battery and chemical reaction optimization (Folch et al., 2023). BO relies on a probabilistic surrogate modeling our belief and uncertainty about the black-box function. One class of surrogates are Gaussian processes (GPs), flexible models giving well-calibrated uncertainty estimates (Rasmussen & Williams, 2005). The most important hyperparameter choice when using a GP is the kernel, which fully defines the model's covariance structure. Popular kernels such as the squared exponential and Matérn kernels are limited to continuous inputs and assume function stationarity.

Tree regression provides a flexible model which is popular due to strong empirical performance, natural regularization and interpretability (Hill et al., 2020; Loh, 2011). Trees allow for mixed inputs and can model non-stationary functions, with hierarchical dependencies. A tree model defines a sequence of domain splits. A single input will go through all the splits until a leaf node is reached which assigns a value to the predictive model. A forest is an ensemble of trees, where each tree is regularized to be a weak learner to avoid overfitting. Despite their performance in regression tasks, tree models have limited application to BO since they lack explicit uncertainty quantification in their predictions, instead relying on empirical variance.

Given an ensemble of $m$ trees, and two inputs $\mathbf{x}$ and $\mathbf{x}'$, we can define a function representing how often the trees agree on the inputs belonging to the same leaf. This notion of *tree agreement* can be used to construct a valid kernel for use in a GP (Balog et al., 2016; Davies & Ghahramani, 2014; Thebelt et al., 2022a). This allows us to use tree models in applications which require well-calibrated estimates such as Bayesian optimization. Tree kernels for model fitting and BO have previously been explored (Balog et al., 2016; Thebelt et al., 2022a), but they have always used a two-step fitting process: first, a tree model is fitted on the data, and *then* the GP is trained using the tree kernel defined by the tree model. While this heuristic gives good results in a variety of benchmarks, it is not principled and hence potentially sub-optimal.

Since the tree structure has an infinite number of parameters, it is at risk of overfitting when optimizing the log-marginal likelihood (Rasmussen & Williams, 2005). To overcome this challenge, we take inspiration from Bayesian tree methods which impose a prior on the tree structure. We obtain a posterior over the tree structure, from which we sample using Markov chain Monte Carlo (MCMC).

We motivate using tree-kernels by exploiting the strong modeling power of tree-based approaches, which can be used for mixed input spaces. Moreover, we can formulate acquisition functions with these tree models as optimization problems that can be solved globally, where gradient-based methods often struggle with discrete inputs.

[1]Imperial College London (London, UK) [2]BASF SE (Ludwigshafen, Germany). Correspondence to: Toby Boyne <t.boyne23@imperial.ac.uk>.

*Proceedings of the 42$^{nd}$ International Conference on Machine Learning*, Vancouver, Canada. PMLR 267, 2025. Copyright 2025 by the author(s).

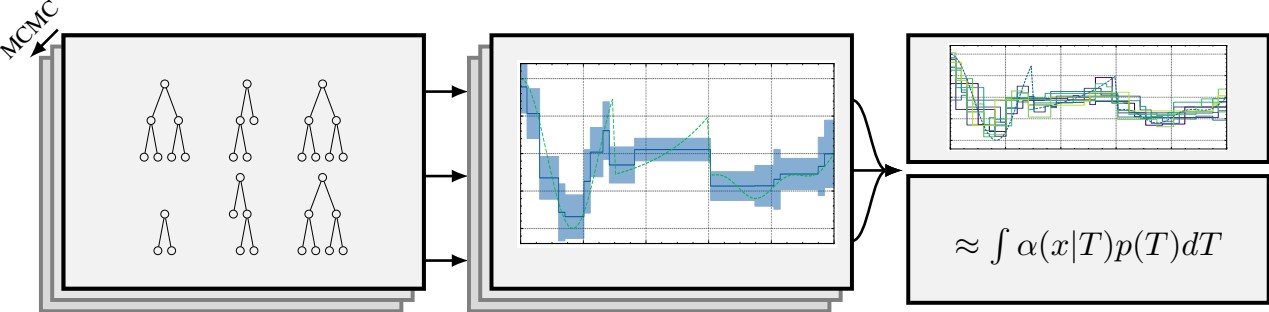

Figure 1: BARK uses MCMC to create samples of forests, each of which define a Gaussian process. We can then use all the GPs to obtain posterior samples of the process, to marginalize over the tree model uncertainty for the acquisition function.

**Contributions.** Our work achieves tree-based GP BO through improved Bayesian modeling. Our main contributions can be summarized as:

- We propose a fully Bayesian method for training tree Gaussian processes through Markov chain Monte Carlo, by exploiting connections between Bayesian tree models and Gaussian processes with tree kernels. We provide a brief summary of our approach in Fig. 1.

- We provide a computationally efficient training algorithm allowing for scalability and accelerating up the Markov chain sampling procedure.

- We provide an empirical analysis of the benefits of our approach, showing its regression ability and achieving state-of-the-art results in a variety of Bayesian optimization benchmarks.

## 2. Related Work

The relationship between Bayesian tree models and GPs has a long history. Chipman et al. (1998; 2010) develop a Bayesian approach for fitting classification and regression trees (BART), which samples a tree structure, and leaf values with a Gaussian prior. Further work by several authors explores using GPs as tree leaf nodes (Gramacy & Lee, 2008; Maia et al., 2024; Wang et al., 2023a), or within other partitions (Luo et al., 2021; 2023). These trees, or other spatial partitioning strategies, allow non-stationary models that may be effective for Bayesian optimization (Eriksson & Jankowiak, 2021; Eriksson et al., 2019; Papenmeier et al., 2022; Wang et al., 2020; Ziomek & Bou Ammar, 2023).

The use of tree models as kernels has been investigated empirically (Thebelt et al., 2022a) and theoretically (Scornet, 2016). Balog et al. (2016) consider tree kernels in the context of Mondrian processes and show that the Mondrian kernel approximates the Laplace kernel while providing a connection to random forests.

Linero (2017) shows that, under certain priors, BART converges to a GP with a Laplace kernel as the number of trees increases. Linero (2017) also mentions that the Laplace kernel GP is sub-optimal in practice compared to the tree model which has a lower computational cost, and better generalization.

Bayesian optimization over mixed feature spaces has been extensively researched in recent years. Techniques include using RKHS embeddings (Buathong et al., 2020), diffusion kernels (Deshwal et al., 2021), multi-armed bandit style optimization (Ru et al., 2020), and a plethora of other methods (Garrido-Merchán & Hernández-Lobato, 2020; Häse et al., 2021; Daxberger et al., 2021; Daulton et al., 2022; Deshwal et al., 2023; Wan et al., 2021) and benchmarks (Dreczkowski et al., 2023). This surge in interest also includes the provision of software packages for mixed feature BO such as SMAC (Lindauer et al., 2022) and scikit-optimize (Head et al., 2018). More specifically, there are also several approaches using tree-based models for black-box optimization (Ammari et al., 2023; Mistry et al., 2021; Thebelt et al., 2021; 2022b). Jenatton et al. (2017) exploit a known tree structure for more efficient BO. Lei et al. (2021) use BART for BO, however, they do not fully exploit the implicit GP structure in the model as we do. Finally, Thebelt et al. (2022a) show how to use GPs with tree kernels for BO with optimal acquisition function optimization, however, they fit the trees using a two-step procedure.

The rest of the paper will continue as follows: Section 3 provides background on tree kernels and Bayesian optimization. Section 4 provides an overview of the BART procedure, laying the groundwork for BARK. Sections 5 and 6 describe our method, highlighting the developments compared to prior work and detailing the motivation behind these improvements, specifically in the context of BO. Section 7 provides empirical results on a suite of synthetic and applied BO benchmarks with mixed feature spaces. Section 8 summarizes our work, addressing the strengths and weaknesses of the method.

# 3. Background

## 3.1. Gaussian process Bayesian optimization

Bayesian optimization (BO) maximizes a black-box function, $\mathbf{x}_* = \arg\max_{\mathbf{x} \in \mathcal{X}} f(\mathbf{x})$, through the sequential and possible noisy querying of the function. BO balances the exploration-exploitation trade-off via an acquisition function, $\alpha$. The next query is $\mathbf{x}_{i+1} = \arg\max_{\mathbf{x} \in \mathcal{X}} \alpha(\mathbf{x}; \theta, \mathcal{D}_i)$, where $\mathcal{D}_i$ represents all the data gathered up to iteration $i$, and $\theta$ are surrogate model parameters. The feature space $\mathcal{X}$ contains all possible experiments, and may contain continuous, integer, or categorical features.

The unknown function $f$ is typically modeled with a Gaussian process (GP) prior. A GP is a probabilistic, nonparametric model over functions, defined by a mean and covariance function (Rasmussen & Williams, 2005). This prior, combined with a likelihood of data observations, induces a posterior over functions. The kernel parameters are often chosen by optimizing the marginal likelihood, however, it also possible to be fully Bayesian and perform inference over the kernel parameters. This cannot be done analytically, and requires posterior samples of kernel parameters to estimate corresponding integrals. For example, we can marginalize over the kernel parameters when optimizing the acquisition function by taking $S$ posterior samples of the parameters, $\{\theta^{(s)}\}_{s=1}^{S}$ (Snoek et al., 2012):

$$
\begin{aligned}
\mathbf{x}_{i+1} &= \arg\max_{\mathbf{x} \in \mathcal{X}} \int_{\theta} \alpha(\mathbf{x}; \theta, \mathcal{D}_i) p(\theta | \mathcal{D}_i)\, \mathrm{d}\theta \\
&\approx \arg\max_{\mathbf{x} \in \mathcal{X}} \frac{1}{S} \sum_{s=1}^{S} \alpha(\mathbf{x}; \theta^{(s)}, \mathcal{D}_i)
\end{aligned} \tag{1}
$$

## 3.2. Forest kernels

The forest kernel uses binary decision trees to define a distribution over piecewise-constant functions. This kernel counts the proportion of its $m$ trees that agree on the leaf node in which a pair of data falls into to define a covariance function. We denote the $t$th tree by $T_t$, which contains $L_t$ leaf nodes. We define the one-hot vector $\phi(\mathbf{x}; T_t)$ where the non-zero entry corresponds to the leaf that contains the datapoint $\mathbf{x}$. This enables the definition of the kernel:

$$
k(\mathbf{x}, \mathbf{x}') = \frac{\sigma_0^2}{m} \sum_{t=1}^{m} \phi(\mathbf{x}; T_t)^{\mathsf{T}} \phi(\mathbf{x}'; T_t), \tag{2}
$$

where $\sigma_0$ is a scale hyperparameter. Eq. (2) produces positive semi-definite matrices, so is a valid PSD kernel (Davies & Ghahramani, 2014). The data $\mathbf{x}$ and $\mathbf{x}'$ can belong to continuous, categorical, or mixed feature spaces, and the kernel is non-stationary. The forest used in the kernel can be trained independently of the GP (as in Davies & Ghahramani (2014), giving a supervised kernel), or fit jointly with the GP, which we propose in our method.

## 3.3. Bayesian Additive Regression Trees

Bayesian Additive Regression Trees (BART) is a fully Bayesian tree model that has seen a lot of empirical success (Chipman et al., 2010). It is formulated as a sum of trees:

$$
y(\mathbf{x}) = \sum_{t=1}^{m} M_t^{\mathsf{T}} \phi(\mathbf{x}; T_t) + \epsilon, \quad \epsilon \sim \mathcal{N}(0, \sigma_y^2) \tag{3}
$$

where $M_t = [\mu_{t1}, \cdots, \mu_{tL_t}] \sim \mathcal{N}(0, \mathbf{I}\sigma_\mu^2/m)$ for some fixed parameter $\sigma_\mu^2$. $T_t$ represents the tree structure and $M_t$ the leaf values. Chipman et al. (2010) show how priors on the tree structure enable Bayesian inference using Gibbs sampling (Gelfand, 2000). The model shares a close implicit relationship with the tree kernel. As noted by Linero (2017), the expectation of the BART posterior over the leaf values $\mathcal{M}$ (conditioned on a set of trees $\mathcal{T}$) is:

$$
\begin{aligned}
\mathbb{E}_{\mathcal{M}} \left[ f(\mathbf{x}) f(\mathbf{x}') \right] &= \sum_{t=1}^{m} \mathbb{E}_{\mathcal{M}} \left[ M_t^{\mathsf{T}} \phi(\mathbf{x}; T_t) \cdot M_t^{\mathsf{T}} \phi(\mathbf{x}'; T_t) \right] \\
&= \frac{\sigma_\mu^2}{m} \sum_{t=1}^{m} \phi(\mathbf{x}; T_t)^{\mathsf{T}} \phi(\mathbf{x}'; T_t)
\end{aligned}
$$
$$\tag{4}$$

Eq. (4) is equivalent to the kernel in Eq. (2), conditioned on $\sigma_0^2 = \sigma_\mu^2$. However, even with the same covariance structure as the tree kernel, all MCMC draws from BART result in deterministic functions, and require many more samples to approximate the distribution that the tree kernel explicitly defines. The effectiveness of BART in BO is therefore limited, since we do not have access to the uncertainty quantification required for building standard acquisition functions.

# 4. The BART model

The BART procedure provides a basis for the BARK sampling mechanism. Here, we present an overview of the method for generating MCMC samples of tree functions - see Chipman et al. (2010) for a comprehensive description.

The BART model consists of three key ingredients: the priors placed on tree functions, the likelihood of observations given a tree function, and the proposals used in sampling the posterior. The priors multiplied by the likelihood of observations induce a posterior $p(\mathcal{T}, \mathcal{M}, \sigma_y^2 | \mathcal{D})$ over the sum-of-tree model. This posterior is intractable, but MCMC sampling can obtain posterior samples.

This MCMC algorithm is largely defined by the Metropolis-Hastings step (Hastings, 1970). Steps are taken in the hyperparameter space according to a transition kernel, $q(\theta \to \theta^*)$, which are rejected/accepted according to the acceptance probability $a$ of the new state. By taking many steps, the samples converges to the posterior over $\theta$. The acceptance

probability is decomposed into three terms,

$$a(\theta, \theta^*) = \min\left(\underbrace{\frac{q(\theta^* \to \theta)}{q(\theta \to \theta^*)}}_{\text{transition}} \cdot \underbrace{\frac{p(\mathbf{y}|\mathbf{X}, \theta^*)}{p(\mathbf{y}|\mathbf{X}, \theta)}}_{\text{likelihood}} \cdot \underbrace{\frac{p(\theta^*)}{p(\theta)}}_{\text{prior}}, 1\right).$$

**BART prior.** The regularization prior defined in the BART model is a prior on the depth of the nodes of a tree. The probability of any given node being a decision node is:

$$\alpha(1 + d)^{-\beta}, \quad \alpha \in (0, 1), \beta \in [0, \infty) \quad (5)$$

The number of trees $m$ in the forest is fixed, e.g., $m = 50$ (Kapelner & Bleich, 2016). To assign a decision rule to the node $\eta$ in the tree, a sample is drawn from the decision rule prior. The feature on which a node is split is drawn uniformly, and the split value is drawn uniformly from the set of unique values in the data that reaches node $\eta$. The prior on the observation noise, parameterized by $(\nu, q)$, is given by $\sigma_y^2 \sim \text{InverseGamma}(\nu/2, \nu t/2)$. For a set of observations with variance $\hat{\sigma}^2$, $t$ is chosen such that $\Pr(\sigma_y^2 < \hat{\sigma}^2) = q$.

**BART likelihood.** The distribution of a set of observations $\mathbf{y}$, under the assumption of Gaussian noise, is

$$\mathbf{y}|\mathbf{X}, \mathcal{T}, \mathcal{M}, \sigma_y \sim \mathcal{N}\left(\sum_{t=1}^{m} M_t^\mathsf{T}\phi(\mathbf{X}; T_t), \sigma_y^2 I\right)$$

The likelihood is conditioned on the sampled leaf values $\mathcal{M}$. Since all other leaf values are fixed when sampling the $j$th tree, the authors compute the likelihood in terms of $\mathbf{R}_j := \mathbf{y} - \sum_{t \in [m] \backslash j} M_t^\mathsf{T}\phi(\mathbf{x}; T_t)$.

**BART proposals.** To explore the posterior of trees, Chipman et al. (2010) define a set of proposals. Each proposal makes a small change to a tree's structure: growing a leaf node by assigning a decision rule from the rule prior; pruning a pair of leaf nodes; changing the rule of a singly internal node (a decision node where both children are leaves). We omit the swap rule in line with other BART implementations (Kapelner & Bleich, 2016; Linero & Yang, 2018).

## 5. The BARK model

### 5.1. Motivating the Bayesian treatment of the kernel

Despite existing work on tree kernels for Gaussian processes, there is still limited literature on tree kernels in a Bayesian optimization setting. We consider two key areas for improvement.

**Two-step fitting.** Tree kernels such as those studied by Davies & Ghahramani (2014) and Thebelt et al. (2022a) fit the GP model in two steps. First, they fit a tree independently, using gradient boosting (Friedman, 2001) or random

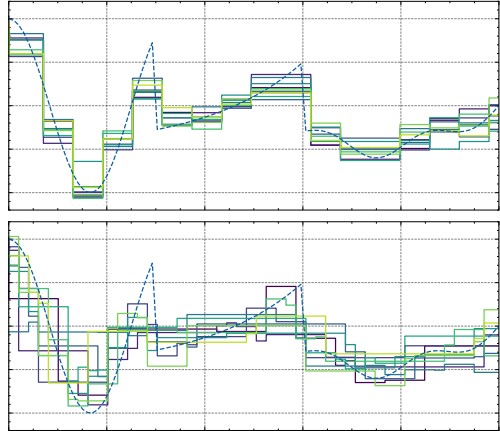

Figure 2: Allowing the tree kernel to vary between samples produces a richer posterior distribution, whilst samples are still piecewise-constant: (top) samples from a GP with a fixed tree kernel, and (bottom) samples from BARK.

forests (Breiman, 2001). Second, they maximize the model likelihood (ML-II) to obtain the noise and scale GP hyperparameters. The tree model is fit using a different objective than the GP, meaning the resulting tree structure may not necessarily maximize the likelihood of the data. Rather than fitting the tree and the GP in two steps, BARK jointly fits the tree kernel and the GP.

**Point estimates of tree posterior.** Some existing tree kernel methods use a one-step algorithm to fit the kernel and treat the noise hyperparameter as fixed, for example (Cohen et al., 2022). However, existing methods only fit a single tree model which is a *maximum a posteriori* estimate. Since the tree kernel is nonparametric, the GP effectively has infinite hyperparameters. The ML-II estimate tends to overfit with many parameters (Rasmussen & Williams, 2005), which can be mitigated by a fully-Bayesian treatment of the kernel (Ober et al., 2021). Moreover, the posterior distribution over functions defined by the model is less rich than the full posterior over tree models, as the tree structure is fixed for all functions sampled from the model, see Fig. 2.

BARK therefore defines a probabilistic model of the tree kernel, and performs MCMC to explore the posterior of trees. We consider the tree model as a nonparametric kernel. The BARK hyperparameters are the parameters defining the forest generating process, i.e. the parameters of the tree priors. These parameters are interpretable, and can be used with default values.

### 5.2. Differences from the BART model

While we take inspiration from BART, relying upon its well established predictive performance, BARK has some key differences from the BART model. We explore these by

considering the three terms of the acceptance probability.

**BARK prior.** In Chipman et al. (2010), the authors normalize observations then place a prior on the variance of the leaves as $\frac{1}{2k}$, where $k$ is to be tuned. We remove the $k$ parameter, instead opting to standardize the outputs and fix the scale $\sigma_0^2 = 1$, which has been shown to work well in Bayesian optimization (Hvarfner et al., 2024).

BARK adopts the same inverse gamma distribution for the noise, $\sigma_y^2$, as BART. Since the data is standardized, the noise prior will be the same for any dataset.

**BARK likelihood** The marginal likelihood of the Gaussian process is computed from $\mathbf{y}$ distributed as

$$\mathbf{y}|\mathbf{X}, \mathcal{T}, \sigma_y \sim \mathcal{N}\left(\mathbf{0}, \frac{1}{m}\sum_{t=1}^{m}\phi(\mathbf{X}; T_t)^\mathsf{T}\phi(\mathbf{X}; T_t) + \sigma_y^2 I\right)$$

Crucially, our kernel perspective means that we never sample leaf values - the marginal likelihood integrates over all the values of the leaves. In this way, BARK samples *distributions*, which can then be used to build acquisition functions, motivating our use of the GP likelihood. In contrast, BART samples the leaf values sequentially, hence uses the likelihood conditional on leaf values. In order to approximate the posterior leaf distribution, BART requires a large number of function samples, which in turn means that it is infeasible to formulate an optimization problem over the BART acquisition function. Both BART and BARK have the same probabilistic model: marginalizing the BART likelihood over leaf values yields the BARK likelihood, as in Eq. (4).

**BARK proposals.** In BART, splitting points are typically uniformly sampled from the data seen by a given node (Chipman et al., 2010; Kapelner & Bleich, 2016). This choice is made such that any leaf node will contain a nonempty subset of the domain (no 'logically empty' leaves). However, this approach performs poorly in the BO setting, since finding potential future queries requires good uncertainty estimation far from data (see Section 6). Instead, we sample the splitting rule uniformly from the domain. To avoid creating logically empty nodes, each splitting rule is sampled from the subspace of the domain that reaches that node. For example, if $x \in [0, 1]$, and the root splitting rule is $x > 0.5$, then the splitting value of the left child is be sampled from $[0, 0.5]$. For categorical features with $S$ categories that reach node a node, the splitting rule is sampled uniformly from $P(S) \setminus \{S, \{\}\}$, i.e., the power set of $S$ excluding trivial splits. This enables greater sample efficiency than a one-hot encoding of categories. We investigate the impact of this modeling choice in Appendix B for toy regression examples.

BARK cannot sample directly from the posterior over $\sigma_y^2$, and so must sample the noise using MCMC. To enforce non-negativity, a proposal is generated by taking a Gaussian

walk in an unconstrained space, using the softplus transform. Details for the BARK proposals are given in Appendix H.

### 5.3. Computational consideration

When we have sampled $N$ data points, the matrix inversion required in the fitting procedure for BARK has $O(N^3)$ complexity. This cubic order is unavoidable due to inverting the covariance matrix when computing the log likelihood in the MCMC steps. However, we can reduce the cost in computing this term for tree proposals by leveraging the low-rank nature of the contributions of individual trees. This is similar to other methods used for fast inverses of tree kernels (Balog et al., 2016; Lee et al., 2015), applied to sequential updates to trees in a forest.

For a dataset $X = [\mathbf{x}^{(1)}, \cdots, \mathbf{x}^{(N)}]^\mathsf{T}$, we define $\Phi \in \{0, 1\}^{(m \times N \times L)}$, where $\Phi_{jnl} = [\phi(\mathbf{x}^{(n)}; T_t)]_l$ and $L = \max_t L_t$. This gives the following expression for the kernel:

$$K_\theta = \frac{\sigma_0}{m}\sum_{j=1}^{m}\Phi_j\Phi_j^\mathsf{T} + \sigma_y^2 I$$

During the MCMC procedure, for a proposal where the $t$th tree is updated, the difference in log-likelihood between the previous GP and the proposed GP can be computed:

$$\log\left(\frac{p(\mathbf{y}|\mathbf{X}, \theta^*)}{p(\mathbf{y}|\mathbf{X}, \theta)}\right) = -\mathbf{y}^\mathsf{T}(K_{\theta^*}^{-1} - K_\theta^{-1})\mathbf{y}$$
$$-(\log|K_\theta^*| - \log|K_\theta|)$$

This requires computing $K_{\theta^*}^{-1}$ and $\log|K_{\theta^*}|$, both of which are $O(N^3)$ operations. However, $K_\theta^{-1}$ and $\log|K_\theta|$ will have already been computed in the previous iteration, and so we can rewrite this as a matrix update,

$$K_{\theta^*} = \frac{\sigma_0^2}{m}\sum_{j\in[m]\setminus\{t\}}\Phi_j\Phi_j^\mathsf{T} + \frac{\sigma_0^2}{m}\Phi_t^*\Phi_t^{*\mathsf{T}} + \sigma_y^2 I$$
$$= K_\theta - \frac{\sigma_0^2}{m}\Phi_t\Phi_t^\mathsf{T} + \frac{\sigma_0^2}{m}\Phi_t^*\Phi_t^{*\mathsf{T}}$$

Since each $\Phi_t$ matrix has rank $L_t$, and $L_t \ll N$ by the depth prior, an update to a tree structure gives two low-rank updates. This can be exploited using the Sherman-Morrison formula (Woodbury, 1950), and the matrix determinant lemma (Harville, 1997), to compute the matrix inverse of the updated kernel matrix in $O(N^2 L_t)$ complexity:

$$(A \pm UU^\mathsf{T})^{-1} = A^{-1}(I - U(U^\mathsf{T}A^{-1}U \pm I)^{-1}U^\mathsf{T}A^{-1})$$
$$\log|A \pm UU^\mathsf{T}| = \log|A| + \log|I \pm U^\mathsf{T}A^{-1}U|$$

## 6. BARK for Bayesian optimization

While BARK can be used for regression, its strength lies in its performance in Bayesian optimization. In this section, we outline how the model has been designed for BO.

## 6.1. Optimizing the acquisition function

Typical BO approaches use gradient-based methods to optimize the acquisition function (AF), such as L-BFGS (Zhu et al., 1997). However, our piecewise constant prior has zero gradient almost everywhere. We turn to mixed integer linear programming to define the optimization problem. We use the framework provided by Thebelt et al. (2022a); Mišić (2020) to encode the tree structure using constraints on binary variables. The full formulation is given in Appendix J. This has the additional benefit of global optimization, where gradient-based methods can be stuck in local optima.

**Integrated acquisition function.** Each sample from the MCMC gives the parameters $(\mathcal{T}, \sigma_y)$, defining a GP kernel and likelihood. A function drawn from BARK is drawn from one of these GPs with equal probability; a BARK predictive distribution is a mixture of Gaussians (Lalchand & Rasmussen, 2020). We can therefore approximate the integrated acquisition function from Eq. (1).

We use the Upper Confidence Bound (UCB), which is a function of the mean, $\mu$, and standard deviation, $\sigma$, of the GP, $\alpha_{\text{UCB}}(\mathbf{x}) = \mu(\mathbf{x}) + \kappa\sigma(\mathbf{x})$. The integrated UCB is a convex function of $\mu$ and $\sigma$, which leads to significant speed increase in solving the global optimization problem. After the optimization is solved and a new datapoint is queried, we sample new tree kernels from the MCMC, using the final sample from each parallel chain of the previous BO iteration as an initialization.

## 6.2. Uniform splitting rule

In Section 5, we describe the new prior over the splitting rules at nodes, where we sample the split values uniformly across the domain. While the original prior worked well in the high data regression setting, we show the importance of this change in the BO setting to encourage exploration.

A key benefit of the change is the better uncertainty quantification. When sampling splits only at datapoints, the uncertainty is constant in the region between datapoints. This no longer has the property that uncertainty increases as the distance from observation increases, an essential ingredient in a well-performing BO surrogate. Moreover, the predictive uncertainty outside the range of observations is constant, leading to poor extrapolation ability. This is demonstrated in Fig. 3; due to the poor uncertainty quantification when splits are sampled at datapoints, the UCB maximizer fails to properly explore the domain.

A further issue with sampling splits from the data is the tendency to cluster queries. Since BO aims to find the optimum value of a function, any exploitative function queries will cluster around known good values. This leads to clusters of observations near the optima. If the splitting rule prior assigns equal splitting probability to each datapoint, then

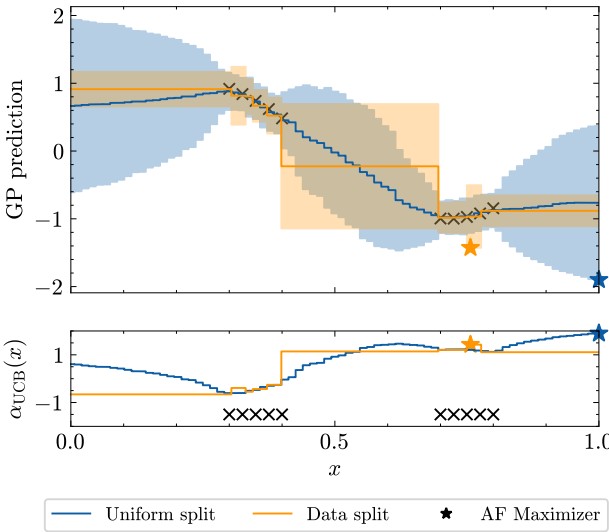

Figure 3: GP predictions and UCB for two BARK GPs on a minimization task: (blue) splits sampled uniformly, and (orange) splits sampled at datapoints. By sampling splits uniformly, the acquisition function (AF) maximizer exhibits better exploration.

the tree structure prior will concentrate on regions that are dense with observations, which leads to over-exploitation.

## 6.3. The Laplace approximation for regret bounds

The functions in the BARK prior are discontinuous and non-stationary. This violates the typical assumptions required for regret bounds in the Bayesian optimization literature (Srinivas et al., 2012). Previous work shows that, under some strong assumptions, the tree kernel can be approximated as a Matern-$\frac{1}{2}$ kernel (Linero, 2017), which in turn admits sub-linear regret bounds of $\tilde{O}(T^{\frac{1+2D}{2+2D}})$, where $D$ is the dimension of the domain (Wang et al., 2023b). Appendix I provides further discussion on the quality of this approximation.

# 7. Experiments

This section describes experiments comparing the regression capabilities of BARK and BART, and evaluates the BARK model against a selection of baselines on a set of synthetic and applied Bayesian optimization benchmarks[1].

**Baselines.** We compare BARK to several baselines. For each baseline, we use the upper confidence bound (UCB) AF. BART uses the Chipman et al. (2010) surrogate, evaluated on a grid, and uses function samples to estimate the UCB (Wilson et al., 2018). GP-RBF uses the squared-exponential

---

[1]The code to run these experiments is available at https://github.com/TobyBoyne/bark.

kernel, with the default priors from Hvarfner et al. (2024). For enumerable mixed feature spaces, we use a linear combination of sum and product kernels, as in Ru et al. (2020). To optimize over mixed feature spaces, we use an alternating approach to perform optimization, similar to (Wan et al., 2021), where we alternate between optimizing the continuous parameters with gradient-based methods, and exploring local perturbations in the discrete parameters. For domains where the number of combinations of categories is low, we also include an enumeration approach, where each unique combination of discrete values is fixed and many parallel continuous optimizations are performed. LeafGP uses a two-step forest kernel, where the forest is fit using gradient boosted trees, then used as a kernel in a GP (Thebelt et al., 2022a). This model can be used in a global optimization of the UCB acquisition function. SMAC uses an ensemble of trees, using sample variance to measure uncertainty (Lindauer et al., 2022). Entmoot defines a mean function with a tree surrogate, and uses a distance-based uncertainty (Thebelt et al., 2021). Finally, BARKPrior samples tree kernels from the prior, without performing the MCMC steps to sample the posterior. Appendix D provides further details of the methods.

## 7.1. Model fit comparison

We perform regression with both BART and BARK. The purpose of this section is to demonstrate that we do not lose predictive power when using the kernel perspective of the BARK model. We show this in the tabular regression setting, where BART is already known to be strong. We use the default setting for the model hyperparameters; for discussion on the selection of these values, see Appendix G.

We use real datasets from the UCI Repository (Cortez & Silva, 2008; Nash et al., 1994; Quinlan, 1993; Yeh, 1998; Dua & Graff, 2017). We compare the predictive performances in Table 1. The performance of the two models is similar, suggesting that BARK can achieve similarly strong regression performance to BART. Our slight increase in NLPD performance may be due to our GP perspective, which effectively allows us to marginalize over the distribution of leaf values. Appendix F discusses the training time on these regression problems, and Table 7 provides a comparison to the GP-RBF model.

We further investigate the trajectory of the marginal log likelihood of the MCMC chain in Fig. 4. As the MCMC process makes local proposals, each sample is highly correlated with the previous sample, and the correlation decreases as the distance between samples (or *lag*) increases. This is quantified by the autocorrelation of the samples $\{X\}_{s=1}^S$, $\rho(\tau) = \mathbb{E}[(X_{t+\tau} - \mu)(X_t - \mu)]/\sigma_X^2$, where $\tau$ is the lag. Since we are limited by the optimization procedure to a small number of samples, we want our samples to be almost

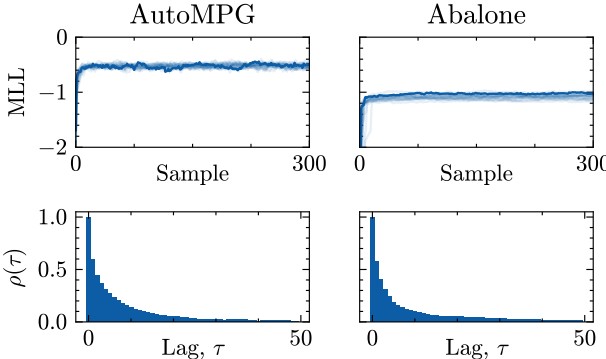

Figure 4: Evolution of the marginal log likelihood (MLL) of the training data over MCMC samples, and the corresponding autocorrelation, $\rho(\tau)$. Lines are plot for 20 MCMC chains, with a single random chain shown in solid colour.

entirely uncorrelated. We find that the samples become uncorrelated after 50 samples, which suggests that the chosen thinning rate of 100 is sufficient.

## 7.2. Synthetic benchmarks

We perform Bayesian optimization on four mixed-space synthetic benchmarks, initialized with $\min(2D, 30)$ datapoints sampled uniformly from the domain. For each method, we report the median and inter-quartile range across 20 runs.

*TreeFunction* and *TreeFunctionCat* are functions sampled from the BART prior, each with 10 continuous dimensions, and the latter with an additional 10 categorical dimensions. *DiscreteAckley* and *DiscreteRosenbrock* are partially discretized functions from Dreczkowski et al. (2023); Bliek et al. (2021). Appendix D has further benchmark details.

We show the results in Fig. 5. A key takeaway here is that, despite *TreeFunction* and *TreeFunctionCat* being drawn from the BART prior, the BART baseline does not outperform LeafGP and BARK. This shows the strength of the optimization formulation, and that a grid-based UCB evaluation is insufficient to find the global optimum. BARK both captures uncertainty over the tree structure, and optimizes the UCB efficiently, to achieve the strongest performance.

*DiscreteRosenbrock* shows a failure case of BARK; the optimum lies in a very shallow region, which the piecewise-constant prior of BARK models worse than the smooth prior of the GP-RBF, and Entmoot's distanced-based uncertainty quantification. These synthetic benchmarks tend to favor the smoothness assumptions of the RBF kernel over the stepwise-constant functions in the BARK posterior. We provide further results for continuous synthetic functions in Appendix E.2.

Table 1: Benchmark performance of the BART model against BARK, given as mean (and standard deviation). The number of training points used for each benchmark is given by $n$. The number of dimensions for each problem is given by $D$ (continuous + integer + categorical). The better performing model is in bold for each metric.

| Benchmark | $n$ | $D$ | NLPD ↓ | | MSE ↓ | |
|---|---|---|---|---|---|---|
| | | | BART | BARK | BART | BARK |
| Abalone | 400 | 7+0+1 | 1.10 (0.06) | **1.09** (0.06) | 0.53 (0.06) | **0.52** (0.06) |
| Auto MPG | 100 | 4+3+0 | 0.46 (0.12) | **0.45** (0.08) | **0.14** (0.02) | 0.15 (0.02) |
| Concrete | 300 | 8+0+0 | 0.54 (0.02) | **0.31** (0.06) | 0.17 (0.01) | **0.11** (0.01) |
| Student Performance | 250 | 0+13+17 | **1.27** (0.09) | **1.27** (0.06) | **0.72** (0.08) | 0.73 (0.07) |

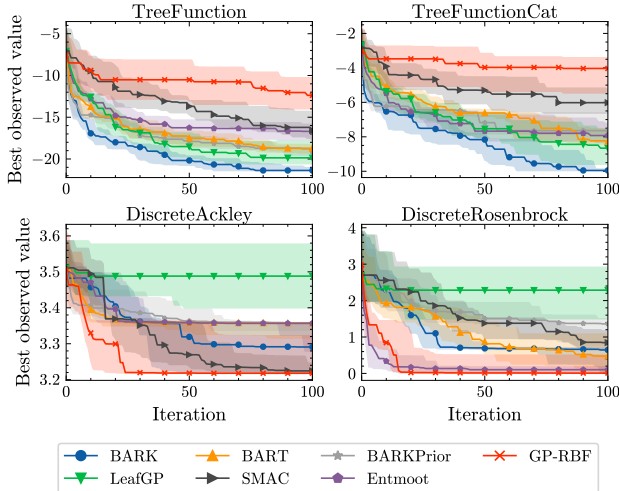

Figure 5: Optimization of synthetic benchmarks. The shaded regions contain the 25th and 75th percentile of regret achieved across the 20 runs.

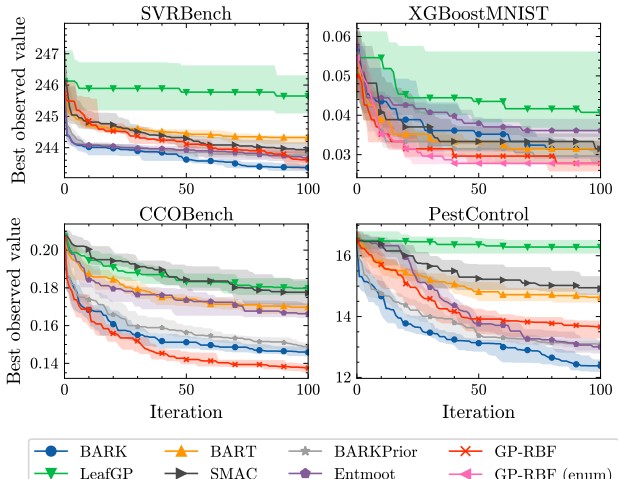

Figure 6: Optimization of applied benchmarks. XGBoostM-NIST allows enumeration of GP-RBF, whereas other benchmarks require alternating optimization.

### 7.3. Applied benchmarks

We evaluate these methods on applied benchmarks. We use the hyperparameter optimization benchmarks *SVRBench* and *XGBoostMNIST* (Dreczkowski et al., 2023). We further evaluate using *CCOBench* (Dreifuerst et al., 2021), which optimizes the configuration of antennas to maximize network coverage, and *PestControl* (Dreczkowski et al., 2023), optimizing the choice of pesticide used at 25 different stations. Fig. 6 gives the results.

BARK outperforms BARKPrior, which itself is a strong method. This shows that the global UCB optimization over tree kernels alone leads to good optimization performance, but proper inference leads to greater improvements. BARK also consistently outperforms LeafGP in both synthetic and applied problems, demonstrating the importance of capturing the uncertainty over tree structure.

GP-RBF achieves the best final value on *XGBoostMNIST*. When it is feasible to enumerate over every categorical com-

bination, GP-RBF is a strong choice. However, in the other applied benchmarks where the alternating optimization approach must be employed, BARK is highly competitive with GP-RBF.

**Bandit optimization with material design.** BART has been used to perform material selection using Bayesian optimization in Lei et al. (2021). Given a dataset of 403 candidate compounds (specifically, a class known as MAX phases), we maximize the bulk modulus and minimize the shear modulus (Talapatra et al., 2018). We present a comparison of BARK's performance on this task, alongside BART and GP-RBF, in Fig. 7.

Since this benchmark has a (small) finite number of candidates, it reveals that BART and BARK have similar BO performance when the acquisition function optimization is ablated. Furthermore, we see that the tree-based methods are better surrogates for this problem than the RBF product kernel, as BARK and BART identify the best candidate in fewer iterations than GP-RBF.

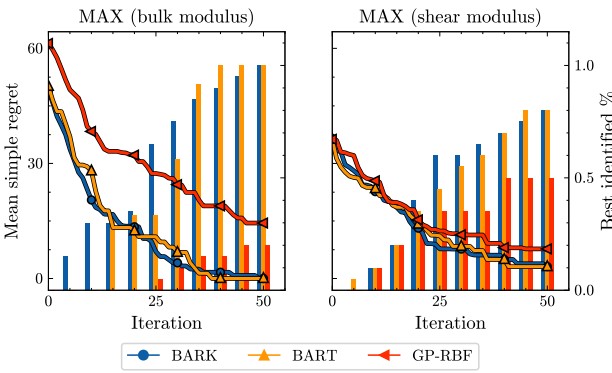

Figure 7: Optimization over compounds in the MAX dataset. We report the mean of the simple regret across 20 runs (lines), and the fractions of the runs that identify the compound with the best objective value (bars).

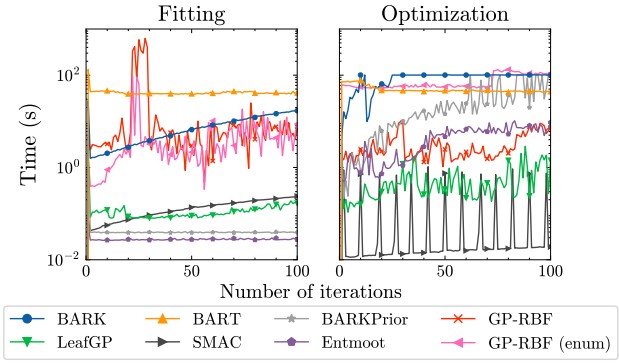

Figure 8: Time taken to fit the surrogate and optimize the acquisition function for the XGBoostMNIST benchmark.

### 7.4. Wall-time discussion

BARK's most significant limitation is its wall-time performance. Fitting the model takes approximately 50 seconds, and the optimization of the acquisition function is limited to 100 seconds. This is due to the combination of the expensive MCMC procedure, and the large optimization formulation. We compare the fitting and optimization times against other methods in Fig. 8, and provide a further selection of wall-time comparisons in Appendix F.2.

We therefore recommend BARK for BO settings where the objective is expensive to evaluate, taking at least several minutes, or having a large associated financial cost. Here, the benefit of an accurate model with strong uncertainty quantification is worth the cost paid in wall-time. Note that the PestControl and MAX (material design) benchmarks reflect examples of such black-box functions. For settings where experiments are cheap and/or function evaluations are quick, we recommend alternate methods.

## 8. Conclusion

We present BARK, a fully Bayesian Gaussian process tree kernel, that jointly trains the tree structure with the GP. This model explores a similar posterior to the additive tree model BART, but provides explicit uncertainty quantification, making BARK a strong surrogate for Bayesian optimization. We show competitive performance on BO, especially in combinatorial settings, due to its strong surrogate and efficient optimizer, beating state-of-the-art approaches.

## Impact statement

This paper presents work whose goal is to advance the field of Machine Learning. There are many potential societal consequences of our work, none which we feel must be specifically highlighted here.

## Acknowledgments

We would like to thank Alexander Larionov for support in implementing benchmarks and baseline methods. Funding for this work was provided by BASF SE, Ludwigshafen am Rhein, and EPSRC through the Modern Statistics and Statistical Machine Learning CDT for TB (EP/Y034813/1) and JPF (EP/S023151/1). TB is also funded by the EPSRC IConIC Prosperity Partnership (EP/X025292/1). RM holds concurrent appointments as a Professor at Imperial and as an Amazon Scholar. This paper describes work performed at Imperial prior to joining Amazon and is not associated with Amazon.

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

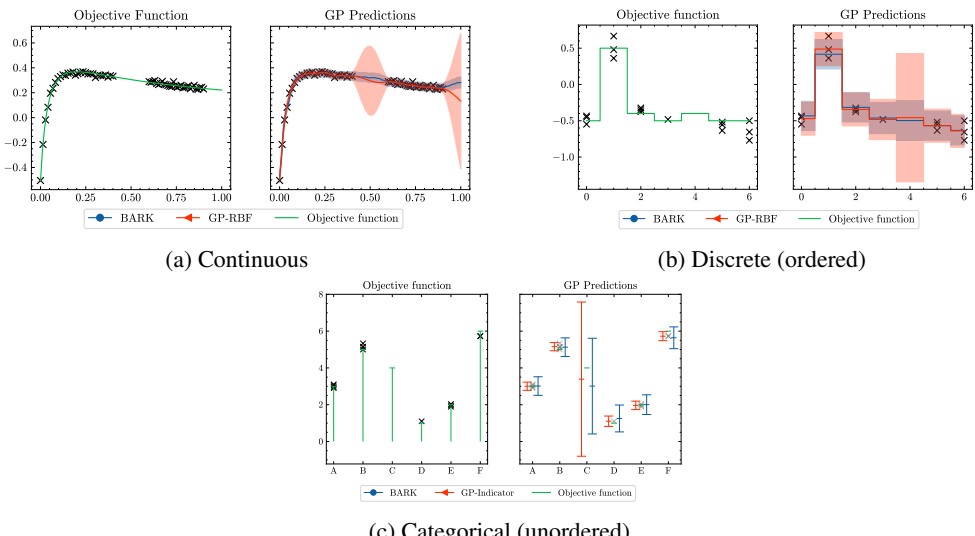

(a) Continuous

(b) Discrete (ordered)

(c) Categorical (unordered)

Figure 9: Regression on a set of 1D toy functions, fit using a BARK GP and an RBF kernel. These functions highlight some notable differences in behavior between the two models.

## A. Related work comparison

We provide a summary of the existing literature in Appendix A, that highlights key similarities and differences between the various BO methods covered in the literature review.

## B. Toy regression examples

As discussed in the main paper, our key motivation behind using tree-based kernels is the optimization of the acquisition function over mixed feature spaces. However, these models also model the underlying function differently, which we investigate in this section by considering some toy 1D problems in Fig. 9.

First, the tree kernel is better able to model non-stationary functions, where the lengthscale varies over the search space. For example, in Figs. 9a and 9b the very short lengthscale for small input values means that the RBF kernel learns a short lengthscale over the whole space, leading to large uncertainty around $x = 0.5$, unlike the BARK model which has a much lower variance in this region.

Second, BARK has a prior belief about the correlation between categorical values. Since each split in the tree structure divides the categorical values into two subsets, the prior probability that two values will be placed in the same leaf is non-zero. Conversely, the indicator kernel used in Ru et al. (2020) explicitly assigns zero correlation between different categorical values. This results in increased variance for unseen categories, as shown in Fig. 9c.

These properties are not necessarily 'better' - different black-box functions require different modeling assumptions. For example, the MAX benchmark in Section 7.3 requires fewer BO iterations when modeled using trees. However, other benchmarks are better modeled with the assumption of smoothness provided by Euclidean GP kernels.

## C. Notation

### C.1. Table of symbols

We list a selection of the key symbols used in this paper in Table 3.

---

[2]This method requires prior knowledge of the structure of the tree function.

Table 2: Comparison of a selection of relevant existing approaches for Bayesian optimization. For each method, we describe the underlying surrogate used to model the objective, and the tool used to optimize the acquisition function. A 'mixture' kernel is the combination of a categorical kernel $k_h$ and a continuous kernel $k_x$ of the form $k(\mathbf{z}, \mathbf{z}') = k_h(\mathbf{h}, \mathbf{h}') \cdot k_x(\mathbf{x}, \mathbf{x}') + k_h(\mathbf{h}, \mathbf{h}') + k_x(\mathbf{x}, \mathbf{x}')$

| Method | Reference | Surrogate | Optimization | GP | Trees/ Partitions | Mixed domain |
|---|---|---|---|---|---|---|
| BART | Chipman et al. (2010) | Bayesian sum-of-trees | Grid | ✗ | ✓ | ✓ |
| ENTMOOT | Thebelt et al. (2021) | Gradient boosted trees | MIP | ✗ | ✓ | ✓ |
| SMAC | Lindauer et al. (2022) | Random forest | Random local search | ✗ | ✓ | ✓ |
| LeafGP | Thebelt et al. (2022) | GP (Tree kernel) | MIP | ✓ | ✓ | ✓ |
| **BARK** | **Ours** | GP (Bayesian tree kernel) | MIP | ✓ | ✓ | ✓ |
| Exact GP | Garrido-Merchán (2020) | GP (Matérn) with input rounding | Gradient descent | ✓ | ✗ | ✓ |
| CASMO-POLITAN | Wan et al. (2021) | Local GP (Mixture) | Alternate gradient descent + discrete perturbation | ✓ | ✓ | ✓ |
| CoCaBo | Ru et al. (2020) | GP (Mixture) | Multi-armed bandit + gradient descent | ✓ | ✗ | ✓ |
| RKHS Embeddings | Buathong et al. (2020) | GP (Deep embeddings) | Genetic optimization using derivatives | ✓ | ✗ | ✓ |
| MiVaBo | Daxberger et al. (2021) | Linear model (with non-linear features) | Thompson sampling | ✗ | ✗ | ✓ |
| Gryffin | Häse et al. (2021) | BNN with simplex projection | Gradient descent | ✗ | ✗ | ✓ |
| HyBO | Deshwal et al. (2021) | GP (Diffusion kernel) | CMA-ES: alternating continuous and discrete subspaces | ✓ | ✗ | ✓ |
| PR | Daulton et al. (2022) | GP (Mixture) | Gradient descent | ✓ | ✗ | ✓ |
| TuRBO | Eriksson et al. (2019) | Local GP (Matérn) | Thompson sampling | ✓ | ✓ | ✗ |
| BAxUS | Papenmeier et al. (2022) | Local GP (Matérn, with embeddings) | Thompson sampling | ✓ | ✓ | ✗ |
| Tree structured | Jenatton et al. (2017) | GP (Matérn) + shared weights | Path-based + gradient descent | ✓ | ✓[2] | ✓ |

Table 3: Symbols used to define tree operations. Any symbol followed by a star, $\cdot^*$, is defined with respect to a proposal

| Symbol | Meaning |
|---|---|
| $m$ | Number of trees in forest |
| $T$ | Tree structure |
| $M$ | Vector of leaf values |
| $\mathcal{T}$ | Forest structure (set of trees, $\{T_t\}_{t=1}^m$) |
| $\mathcal{M}$ | Forest leaf values (set of values, $\{M_t\}_{t=1}^m$) |
| $q(\cdot \rightarrow \cdot^*)$ | Proposal probability |
| $w_0$ | Number of leaf nodes |
| $w_1$ | Number of singly-internal decision nodes (both children are leaves) |
| $\eta$ | Node |
| $d_\eta$ | Depth of node $\eta$ |
| $(\alpha, \beta)$ | Node depth prior parameters |
| $(\nu, q)$ | Noise prior parameters |

## C.2. Model hyperparameters

For clarity, we provide an overview of the parameters used in the BARK model in Appendix C.2.

Table 4: Hyperparameters and parameters of the BARK model. For fixed parameters, their value is given. Parameters that are sampled are given the label *Sampled*.

| Symbol | Description | Value |
|---|---|---|
| $m$ | Number of trees in ensemble | 50 |
| $(\alpha, \beta)$ | Parameters of node depth prior | $(0.95, 2)$ |
| $(\nu, q)$ | Parameters of noise prior | $(3, 0.9)$ |
| $\sigma_0^2$ | Signal variance | 1 |
| $\sigma_y^2$ | Noise variance | *Sampled* |
| $T_i$ | The $i$th tree structure | *Sampled* |

## D. Experimental details

### D.1. Method details

**BARK.** For both model fitting and BO, we use 16 samples for the BARK kernel. Increasing the number of samples would (marginally) improve the model fit, however this would increase the complexity of the optimization problem, meaning that the optimization would take longer to converge. BARK uses 1000 burn-in samples and 400 kernel samples, running in parallel with 4 chains. We use a thinning rate of 100 to obtain the 16 samples. We evaluate the effect of increasing the number of samples in Appendix G.

In BO we only generate the burn-in samples in the first iteration. Since each iteration only adds a single datapoint, the posterior from the previous iteration will be close enough to that of the current iteration that further mixing is not required.

For BARK (as well as BART), we use the default number of trees $m = 50$, as suggested by Kapelner & Bleich (2016), and the default parametrization of the noise, $(\nu, q) = (3, 0.9)$.

**BART.** For model fitting and BO, BART takes 1000 samples to burn-in the MCMC chain, and 1000 posterior samples. This is run in parallel for 4 chains.

There is no procedure in the literature for optimizing acquisition functions for BART, and the MIP formulation would be too

expensive for the 1000s of BART samples required. We therefore evaluate the surrogate on a grid of $\max(2^{5D}, 2^{14})$ points using a space-filling approach (Sobol, 1967). At each iteration, we sample a new grid to better explore the domain. This grid size was chosen to match the wall clock time of the BARK optimization (as shown in Fig. 11).

Each sample $f^{(s)}$ in BART is a single, deterministic function draw, whereas a BARK sample $f_{\text{BARK}}|\mathcal{T} \sim \mathcal{GP}(\cdot, \cdot)$ is a distribution over functions. Therefore, where BARK allows access to the exact UCB for each sample as detailed in Eq. (1), BART requires the sample-based estimation of Wilson et al. (2018), $UCB(\mathbf{x}) \approx \frac{1}{S} \sum_{s=1}^{S} \mu + \kappa \sqrt{\pi/2}|y^{(s)} - \mu|$, where $\mu$ is the empirical average of the observations.

We use the PyMC-BART v0.8.2 implementation (Quiroga et al., 2022).

**GP-RBF.** We use the default priors from (Hvarfner et al., 2024), namely a squared exponential kernel with a log-normal lengthscale prior

$$p(\ell) = \mathcal{LN}(\sqrt{2} + \log \sqrt{D}, \sqrt{3})$$

We use the BoFire v0.0.16 implementation (Dürholt et al., 2024), which in turn uses BoTorch v0.11.3 (Balandat et al., 2020). To optimize mixed space acquisition functions, we use the `optimize_acqf_mixed_alternating` function provided by BoTorch.

**LeafGP.** We use the implementation provided by Thebelt et al. (2022a).

**SMAC.** We use the SMAC3 v.2.0 implementation (Lindauer et al., 2022), using the ask-tell interface.

**Entmoot.** We use the BoFire v0.0.16 implementation (Dürholt et al., 2024).

### D.2. Benchmark details

A description of the benchmarks used in Bayesian optimization is given in Tables 5 and 6. For each BO benchmark, we generate a set of $\min(2D, 30)$ initial points, and run the optimization loop for 100 iterations, reporting the minimum value observed up to iteration $t$.

The experiments were run on a High Performance Computing cluster, equipped with AMD EPYC 7742 processors, with each core allocated 16GB of RAM. For models capable of multithreading, we use 8 CPUs; otherwise, only 1.

Table 5: Synthetic benchmarks for Bayesian optimization. For each benchmark, we give the number of dimensions $D$, and the bounds for the continuous ($\mathbf{x}$), discrete ordinal ($\mathbf{i}$), and categorical ($\mathbf{h}$) features.

| Benchmark | $D$ | Features |
|---|---|---|
| TreeFunction | 10 | $\mathbf{x} \in [0,1]^{10}$ |
| TreeFunctionCat | 20 | $\mathbf{x} \in [0,1]^{10}$ |
| | | $\mathbf{h} \in \{1, \cdots, 5\}^{10}$ |
| DiscreteAckley | 13 | $\mathbf{x} \in [-1,1]^3$ |
| | | $\mathbf{i} \in \{-1,1\}^{10}$ |
| DiscreteRosenbrock | 10 | $\mathbf{x} \in [-5,10]^4$ |
| | | $\mathbf{i} \in \{-5,0,5,10\}^6$ |

## E. Additional results

### E.1. Regression comparison with GP-RBF

In Section 7.1, we show that BART and BARK have similar modeling abilities, and that our kernel perspective on BART still leads to strong regression performance. For completeness, we also provide a comparison to the GP-RBF model, with linear combination of sum and product kernels, in Table 7.

### E.2. Continuous benchmarks

In the main experiments of the paper, we note the motivating use case for BARK is in optimizing over mixed feature spaces. However, we also provide two additional benchmarks here, for continuous synthetic problems. Specifically, we

Table 6: Applied benchmarks for Bayesian optimization. The descriptions of SVRBench, XGBoostMNIST, and PestControl are from the source paper, Dreczkowski et al. (2023).

| Benchmark | $D$ | Features |
|---|---|---|
| SVRBench | 53 | $x_1$ is $\log \epsilon$, $x_2$ is $\log C$, $x_3$ is $\log \gamma$, $\mathbf{h}$ are features to include. $x_1 \in [-2, 0]$ $x_2 \in [-2, 2]$ $x_3 \in [-1, 1]$ $\mathbf{h} \in \{\text{exclude}, \text{include}\}^{50}$ |
| XGBoostMNIST | 8 | $x_1$ is log learning rate, $x_2$ is min split loss, $c_3$ is min split loss, $x_4$ is reg lambda, $i_1$ is max depth, $h_1$ is booster, $h_2$ is grow policy, $h_3$ is objective $x_1 \in [-5, 0]$ $x_2 \in [0, 10]$ $x_3 \in [0.001, 1]$ $x_4 \in [0, 5]$ $i_1 \in \{1, \cdots, 10\}$ $h_1 \in \{\text{gbtree}, \text{dart}\}$ $h_2 \in \{\text{depthwise}, \text{lossguide}\}$ $h_3 \in \{\text{multi:softmax}, \text{multi:softprob}\}$ |
| CCOBench | 30 | $\mathbf{x}$ are transmission powers, $\mathbf{i}$ are downtilts. $\mathbf{x} \in [30, 50]^{15}$ $\mathbf{i} \in \{0, \cdots, 5\}^{15}$ |
| PestControl | 25 | $\mathbf{h}$ are pesticide choices at each of 25 stages. $\mathbf{h} \in \{\text{pest1}, \text{pest2}, \text{pest3}, \text{pest4}, \text{none}\}^{25}$ |
| MAX | 28 | $h_0, h_1, h_2$ are the M, A, and X elements respectively. $\mathbf{x}_{\{1, \cdots, 12\}}$ are chemical properties of the MAX compound. $\mathbf{x}_{\{13, \cdots, 28\}}$ are nuisance features (uniform noise). $h_0 \in M_{\text{elements}}, |M_{\text{elements}}| = 9$ $h_1 \in A_{\text{elements}}, |A_{\text{elements}}| = 12$ $h_2 \in X_{\text{elements}}, |X_{\text{elements}}| = 2$ $\mathbf{x}_{\{13, \cdots, 28\}} \in [-1, 1]^{16}$ |

use the Hartmann and Styblinksi-Tang problems from the SFU library (Surjanovic & Bingham, 2013). We show the BO performance in Fig. 10. The smooth functions are more suited to GP-RBF, however BARK outperforms all other tree-based methods, and still shows good performance in this unfavorable setting on the Styblinski-Tang benchmark.

## F. Time comparison

### F.1. Posterior sampling

BART and BARK both take significantly longer to train than the other baselines, as they are MCMC-based methods. Given $N$ observations, training BART has $O(N)$ complexity, whereas BARK trains with $O(N^3)$ complexity (which is reduced to $O(N^2)$ when sampling tree structures due to the low-rank update discussed in Section 5). We therefore expect BART to scale far more favorably with many observations. In Table 8, we show the time taken to train the models on the regression problems from Section 7. The fitting time for BARK is reasonable in the low-data regime of BO, matching that of BART when only 40 training points are used. Since we apply our model in the BO setting, where the evaluation of the black-box function may take hours, or possibly even days, we argue that this time taken is not a significant limitation of the model. We also note that we use fewer samples for BARK (400 samples per parallel chain) compared to BART (1000 samples per chain).

Table 7: Regression performance of the BART and GP-RBF models, given as mean (and standard deviation). The number of training points used for each benchmark is given by $n$. The number of dimensions for each problem is given by $D$ (continuous + integer + categorical). The better performing model is in bold for each metric.

| Benchmark | $n$ | $D$ | NLPD ↓ | | MSE ↓ | |
|---|---|---|---|---|---|---|
| | | | BARK | GP-RBF | BARK | GP-RBF |
| Abalone | 400 | 7+0+1 | 1.09 (0.06) | **1.06** (0.05) | 0.52 (0.06) | **0.49** (0.05) |
| Auto MPG | 100 | 4+3+0 | **0.45** (0.08) | 0.50 (0.22) | **0.15** (0.02) | **0.15** (0.03) |
| Concrete Compressive Strength | 300 | 8+0+0 | **0.31** (0.06) | 0.39 (0.04) | **0.11** (0.01) | 0.15 (0.01) |
| Student Performance | 250 | 0+13+17 | **1.27** (0.06) | 1.43 (0.06) | **0.73** (0.07) | 1.00 (0.10) |

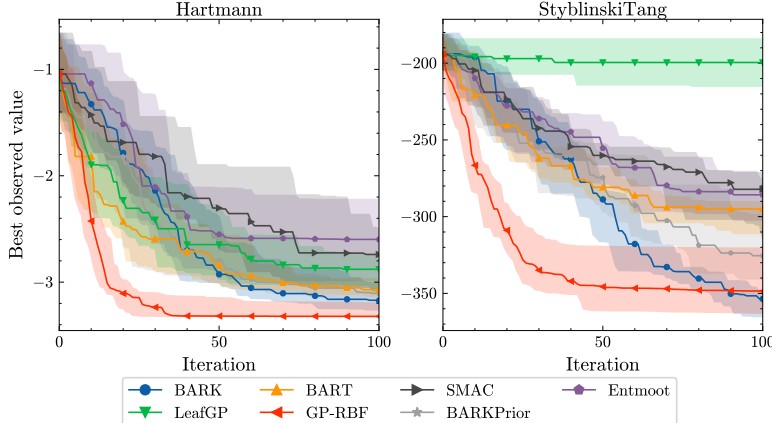

Figure 10: Bayesian optimization performance on benchmarks with a continuous-only domain: Hartmann (6D) and Styblinski-Tang (10D), both from the SFU library of synthetic functions.

Table 8: Mean (and standard deviation) time to generate 1000 samples from the posterior for the BART and BARK models across 20 random data splits. We compare against BART using the PyMC-BART implementation.

| Benchmark | $n$ | Sampling time (seconds per 1000 samples) | |
|---|---|---|---|
| | | BART | BARK |
| Student Performance | 40 | 20.84 (5.86) | 19.36 (7.80) |
| Auto MPG | 100 | 18.09 (6.08) | 71.04 (7.97) |
| Abalone | 400 | 20.81 (4.98) | 951.91 (14.28) |

### F.2. Optimization

We provide a comparison of the time taken to fit and optimize the models in the BO loop in Fig. 11.

BARK is slower than competing methods, taking approximately 50s to fit the model, and 100s to optimize. This is due to the combination of the expensive MCMC procedure, and the large optimization formulation. This is comparable to the fitting time for BART, and the time taken to evaluate BART on a grid of $2^{14}$ points. BARK is best applied in BO settings where the objective is expensive to evaluate (e.g. taking at least several minutes, or having a large associated financial cost). Note that the PestControl and MAX (material design) benchmarks reflect examples of such black-box functions. For settings where experiments are cheap and/or function evaluations are quick, we recommend alternate methods.

## G. Hyperparameter sensitivity

### G.1. Node depth prior

Throughout our experiments, we use the default BART values of $(\alpha, \beta) = (0.95, 2.0)$ for the node depth prior. These are the only hyperparameters for the tree generating process, so it is natural to ask how sensitive our method is to the selection of these values.

We evaluate the regression performance for a grid of $(\alpha, \beta)$ values in Fig. 12. We show that the default values have generally strong performance across both small- and large-data regimes. Furthermore, changing the hyperparameters do not have a significant impact on the observed NLPD - all of the results are well within the standard deviation reported in Table 1. We therefore conclude that our method is largely insensitive to the hyperparameter choice, and use the default BART values.

We also note that, while a smaller value of $\beta$ may lead to stronger regression in some cases, concentrating the prior on deeper trees will make the optimization problem more difficult. Maia et al. (2024) explore a similar sensitivity analysis for their BART-based model, and also opt for the default BART hyperparameters.

### G.2. Number of samples

In all experiments with BARK, we use 16 samples from the posterior - 4 samples collected from 4 parallel MCMC chains. This is a compromise between model fit, and optimization speed: increasing the number of samples improves the performance of the surrogate, at the cost of adding more constraints to the optimization formulation, rendering the task of optimizing the acquisition function more expensive.

In Fig. 13, we show the effect of increasing the number of samples on regression performance. We observe that increasing the number of samples 16 gives marginal improvement on predictive performance, while rendering the optimization problem significantly more difficult.

## H. Posterior sampling

Many of the derivations are included in Kapelner & Bleich (2016). We include all of these terms here for completeness, however derivations are omitted where they are presented in the literature. A table of symbols is provided in Appendix C.

The grow and prune proposals are the inverse of each other, and the change operation is its own inverse. As long as the probability of selecting each proposal is equal to the probability of its inverse proposal, the ratio of proposal probabilities (e.g., the probability of selecting the grow proposal) does not need to be included in the Metropolis Hastings ratio.

**Transition ratio (prune and grow)**    Below is the ratio for the grow proposal. The probability of selecting the target node is $\frac{1}{w_0}$, and the probability of selecting the target node for a prune (the inverse proposal) is $\frac{1}{w_1^*}$. The ratio for the prune proposal is the inverse of this ratio.

$$\frac{q(T^* \to T)}{q(T \to T^*)} = \frac{w_0 \cdot \Pr(\text{rule})}{w_1^*} \tag{6}$$

**Prior ratio (prune and grow)**    The grow prior ratio is given below, and the prune prior ratio is the inverse of this expression.

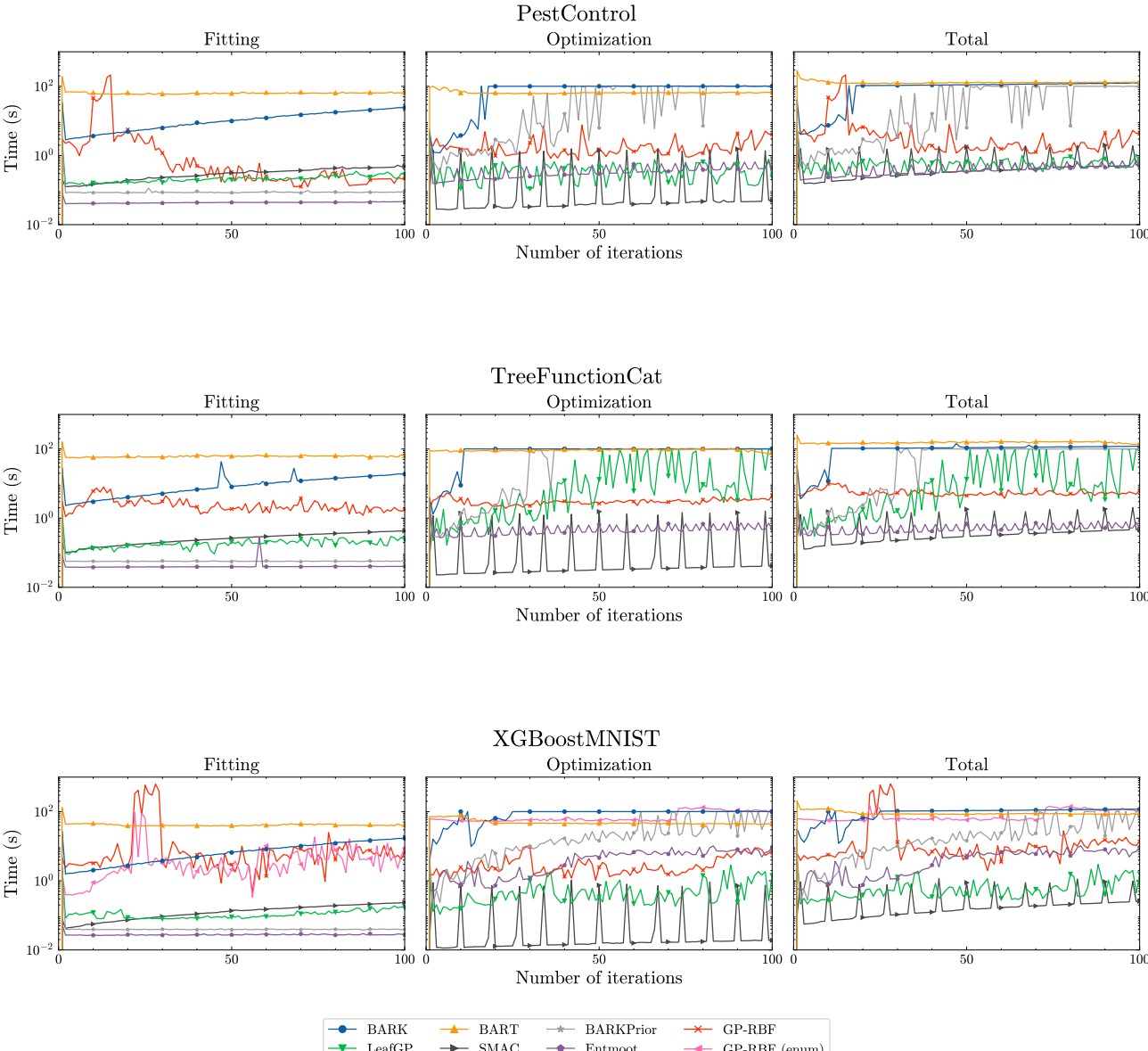

Figure 11: Time taken to fit the surrogate, optimize the acquisition, and combined time for each method used in the Bayesian optimization experiments. We compare the time taken per iteration across three benchmarks. For the XGBoostMNIST benchmark, GB-RBF is optimized by enumerating every discrete combination, leading to increased optimization times in line with BARK.

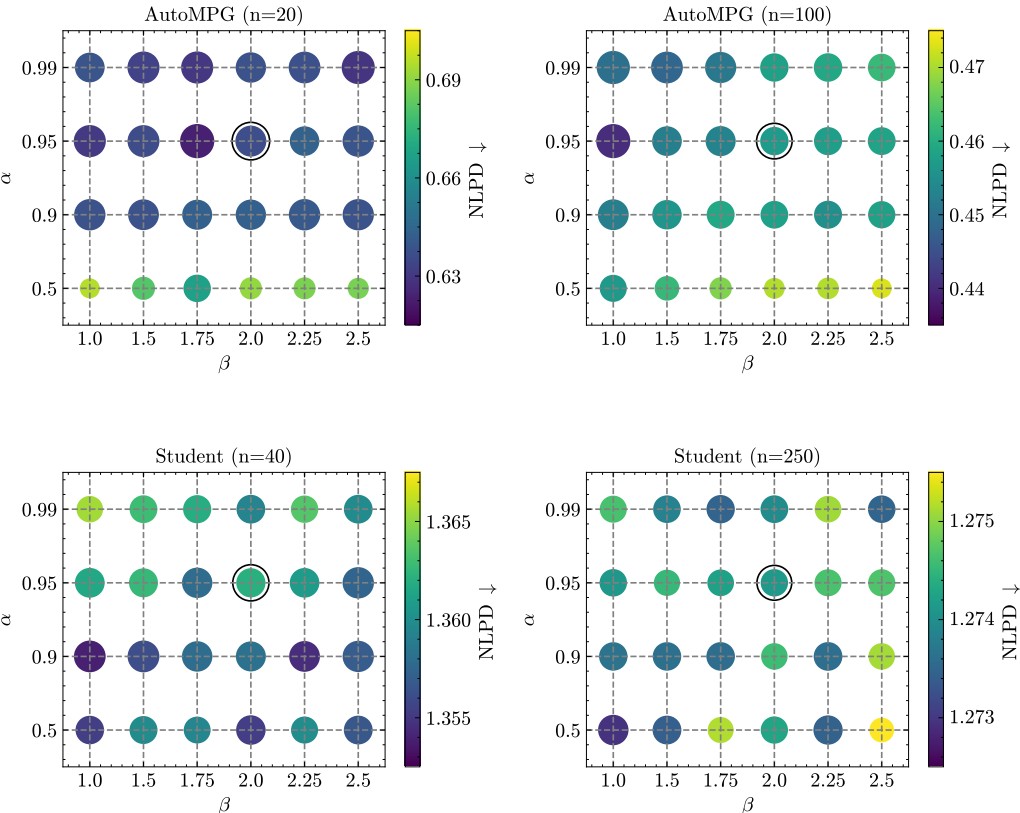

Figure 12: Regression performance of BARK for four tasks, varying the node depth hyperparameters, averaged across 20 test/train splits. Larger, darker dots show stronger performance. The default value pair $(0.95, 2.0)$ is circled. Note that the range of NLPDs for each task is small, and is far less than the standard deviations of performance given in Table 1.

$$\frac{p(T^*)}{p(T)} = \alpha \frac{\left(1 - \frac{\alpha}{(2+d_\eta)^\beta}\right)^2}{\left((1+d_\eta)^\beta - \alpha\right)\Pr(\text{rule})} \tag{7}$$

We note that the splitting rule probability terms in the transition ratio and the prior ratio cancel. This means that, as long as the same distribution is used for the transition kernel as the prior distribution on trees, the acceptance probability is independent of the choice of splitting rule distribution.

**Transition ratio and prior ratio (change)**     As noted in Kapelner & Bleich (2016), the transition ratio and the prior ratio cancel in the expression for the acceptance probability. All that remains is the likelihood term.

**Prior ratio (noise)**     This can be simply calculated using the ratio of the priors placed over the hyperparameter.

**Transition ratio (noise)**     As stated in Section 5, we sample the noise hyperparameter in a transformed space, to ensure that the parameter remains positive. Specifically:

$$\theta = g^{-1}(\sigma^2)$$
$$\theta^* \sim \mathcal{N}(\theta^*; \theta, \sigma_\epsilon^2)$$
$$\sigma^{2*} = g(\theta^*)$$

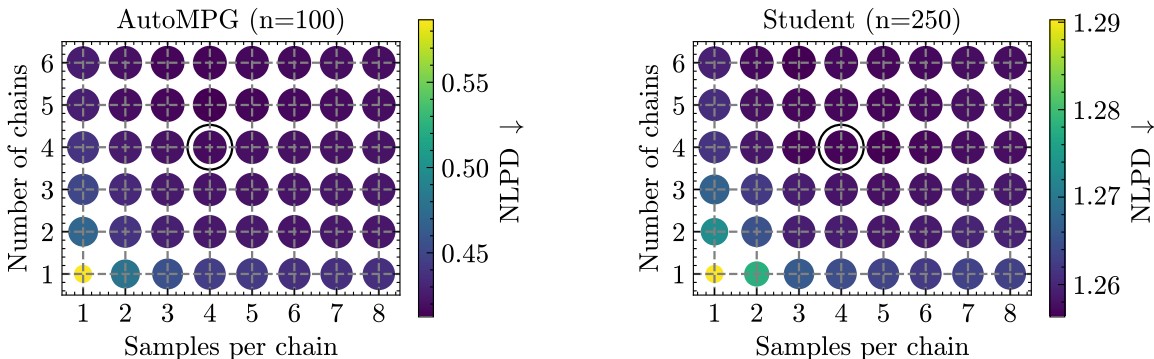

Figure 13: NLPD for varying the number of chains and number of samples per chain, averaged across 10 runs. Larger, darker dots show stronger performance. The BARK default (4, 4) is circled.

where $g : \mathbb{R} \to \mathbb{R}^+$. We use the softplus transform, $g(\theta) = \log(1 + \exp(\theta))$. Given this transform, we can then obtain the probability density function of $\sigma^{2^*}$,

$$f_{\sigma^{2*}}(\sigma^{2*}) = f_{\theta^*}(g^{-1}(\sigma^{2*})) \left| \frac{d}{d\theta^*} g^{-1}(\sigma^{2*}) \right|$$

$$\log f_{\sigma^{2*}}(\sigma^{2*}) = -\frac{1}{2\sigma_\epsilon^2} \left( g^{-1}(\sigma^{2*}) - g^{-1}(\sigma^2) \right)^2$$

$$- \log\left(1 - \exp\left(-\sigma^{2*}\right)\right) + \text{const}$$

We therefore obtain the log-transition ratio,

$$\log \frac{f_{\sigma^{2*}}}{f_{\sigma^2}} = -\left(1 + \frac{1}{\sigma_\epsilon^2}\right) \log \left( \frac{\exp(\sigma^{2*}) - 1}{\exp(\sigma^2) - 1} \right) + \sigma^{2*} - \sigma^2$$

## I. Regret bounds and kernel approximation

Bayesian optimization literature often provides bounds on the cumulative regret during the optimization process. In this section, we discuss why this is a challenge for our method, and provide some discussion on how a regret bound might be attained.

### I.1. General regret bounds: existing methods are not applicable

An initial exploration of the literature in Bayesian optimization and bandits gave us two promising avenues for proving regret bounds, however upon more detailed examination we found the problem to be far more difficult than we originally envisaged – even under strong simplifying assumptions such as the tree structure being known.

The first attempt relates to the direct application of Theorem 1 in Srinivas et al. (2012). Indeed, if we assume the tree structure was known, the kernel is well defined and the bounds would seemingly follow. However, the proof heavily relies on Lipschitz-continuity of the function, and the assumption of an RHKS norm implying differentiability. In our setting, this is not necessarily the case - near decision boundaries, there are discontinuities that cannot be bounded.

An additional avenue that could be explored relates to the bandit literature. Indeed, assume that the objective function, $f$, is sampled from a tree-kernel GP. The complexity of $f$ is bounded: this can be achieved by thresholding the prior probability of the tree structure. Observations of $f$ have Gaussian noise, i.e. $y(\mathbf{x}) = f(\mathbf{x}) + \epsilon$. Even though we have a continuous space, there is a finite number of splits, so the resulting problem is trying to find the best subspace - defining a multi-armed bandit problem. However, the number of resulting arms will be extremely large, therefore standard bounds for independent bandits under UCB will be far too loose.

We could further reformulate the problem in a semi-bandit setting, where each of the trees in the forest is a bandit, and instead of observing individual rewards, we observe the sum of the bandits in so called "super-arms". This setting is explored in the combinatorial multi-armed bandit literature, where there is a collection of $M$ super-arms each with $n$ arms whose sum we can observe. However, in the literature $M << n$, but in our case we have far more splits than trees, so $M >> n$, which again results in very loose bounds. This approach would also be conditioned on a known tree structure, which would not match the setting where the tree structure is being explored.

### I.2. Asymptotic bounds: kernel approximation fails

#### I.2.1. THE LAPLACE KERNEL AS AN APPROXIMATION

The connection between asymptotic tree kernels and the Matern-$\frac{1}{2}$ kernel (often referred to as the Laplace kernel) have been explored in the literature since Balog et al. (2016). This opens up the idea that Matern regret bounds (Wang et al., 2023b) can be combined with approximation guarantees (e.g. Xie et al. (2024)) to obtain asymptotic regret bounds (as the number of trees increases). However, as we will show, further inspection reveals the kernel approximation is too crude for the BART and BARK models to be meaningful.

Firstly, let us begin by showcasing the scenario in which the kernel is a good approximation, as it was originally explored in Linero (2017). We work in the domain $\mathcal{X} = [0, 1]^D$, and in the following setting. Something that is important to note is that the following two assumptions *do not* hold for either BART or BARK models:

**Assumptions.**

1. For any leaf node $\eta \in \mathcal{L}_T$, the depth of the node $d_\eta \sim \text{Poisson}(\lambda)$

2. For any decision node $\eta \in \mathcal{J}_T$, the selected feature $j_\eta \sim \text{Categorical}\left(\frac{1}{D}, \cdots \frac{1}{D}\right)$, and the decision rule $C_\eta \sim \mathcal{U}(0, 1)$.

**Theorem.** The limit of the kernel as $m \to \infty$ is given by

$$k(\mathbf{x}, \mathbf{x}') = \exp\left(-\frac{\lambda}{D}|\mathbf{x} - \mathbf{x}'|_1\right) \tag{8}$$

**Proof.** We aim to find $k(\mathbf{x}, \mathbf{x}')$, which is equal to $1 - \Pr(\text{split} \in [\mathbf{x}, \mathbf{x}'])$. Let $\delta = |\mathbf{x} - \mathbf{x}'|_1/D$. For $n$ splits, the probability of no splits in this interval is $p_n = (1 - \delta)^n$. For the leaf node $\eta$ that contains the point $\mathbf{x}$, let $d_\eta \sim q$. By assumption, the probability mass function of $q$ is the Poisson distribution. Let $\mathbf{x} \sim \mathbf{x}'$ denote the event that $\mathbf{x}$ and $\mathbf{x}'$ fall in the same leaf, i.e., $\Pr(\mathbf{x} \sim \mathbf{x}') = \mathbb{E}_T[\phi(\mathbf{x}, T)^\mathsf{T}\phi(\mathbf{x}', T)]$. Then:

$$\begin{aligned}
\Pr(\mathbf{x} \sim \mathbf{x}') &= \sum_{n=0}^{\infty} q_n p_n \\
&= \sum_{n=0}^{\infty} q_n (1 - \delta)^n \\
&= \sum_{n=0}^{\infty} \frac{\lambda^n \exp(-\lambda)}{n!} (1 - \delta)^n \\
&= \exp(-\lambda) \sum_{n=0}^{\infty} \frac{[\lambda(1 - \delta)]^n}{n!} \\
&= \exp(-\lambda) \exp(\lambda(1 - \delta)) \\
&= \exp\left(-\frac{\lambda}{D}|\mathbf{x} - \mathbf{x}'|_1\right)
\end{aligned}$$

where we use the Taylor expansion of $\exp(\lambda(1 - \delta))$ to obtain the approximation.

The $\frac{1}{D}$ term in the Laplace approximation points to a useful natural property of the BARK kernel: as the dimensionality of the problem increases, the effective lengthscale increases. This leads to a similar behaviour as Hvarfner et al. (2024), where we assume less complex models in higher dimension to mitigate the curse of dimensionality.

If the BARK kernel can indeed be approximated by the Laplace kernel, then we can use the result of Wang et al. (2023b) to show that the cumulative regret is bounded by $\tilde{O}(T^{\frac{1+2D}{2+2D}})$, enjoying sub-linear regret. Even if the approximation is only correct up to some error $\epsilon$, then we could employ Xie et al. (2024) to correct the regret bounds.

### I.2.2. DROPPING SPLIT INDEPENDENCE AND POISSON ASSUMPTIONS

In the previous subsection, we assume that the splits chosen at each decision node are independent. This results in 'logically empty' leaf nodes that cannot be reached by any point in the domain. However, this assumption is necessary to obtain a stationary kernel.

We also note that the assumption of the Poisson distribution of node depths does not follow the BART (nor BARK) prior. The Poisson distribution is obtained by considering the successes of several independent identically distributed events, whereas the probability of a split depends on the depth of the node. The Poisson distribution places too high a probability mass on large trees, which is in practice avoided by the BART depth prior. Moreover, the Poisson distribution is placed on the depth of a given node. However, for any distribution of node depths, a datapoint is more likely to fall into a more shallow node, as the volume of the subset of the domain defined by that node will be larger. This assumption therefore places an even higher probability on a given datapoint falling into a deep node than is observed in practice.

This discrepancy was considered in Petrillo (2024), who show that the GP limit of BART follows a complicated kernel that can needs to be calculated recursively. In this subsection, we alter the assumptions from Linero (2017) and show the approximation is too loose to be used for regret bounds. In particular we consider the following:

1. The splitting rules are sampled uniformly in the subspace defined by previous splits (i.e. there are no logically empty splits).

2. We use the true depth prior used by BART and BARK (i.e. we no longer assume the distribution to be Poisson).

We consider the 'chopping' process. The interval $[0, 1]$ is repeatedly chopped, discarding the region to the right of the chop. The location of each chop is uniformly distributed on the remaining interval: such that $T_d \sim \mathcal{U}[0, 1]$, and the length of the remaining interval $l_d = \prod_{i=1}^{d} t_i$. For a point $x \in [0, 1]$, let the event $S_d$ be the event that $l_d \in [0, x]$ and $l_{d-1} \notin [0, x]$- that is, the $d$th chop disconnects $x$ from 0. This is equivalent to the event that at a split of depth $d$ at a decision node, the points 0 and $x$ are placed in different leaf nodes.

$$T_d \sim \mathcal{U}[0, 1]$$
$$-\log T_d \sim \mathrm{Exp}(1)$$
$$-\log L_d = -\sum_i \log T_i \sim \mathrm{Gamma}(d, 1)$$

Given the distribution of $-\log L_d$, we can compute the cumulative distribution function of $L_d$ as below:

$$F_{L_d}(l_d) = \frac{1}{(d-1)!} \int_0^{\log l_d} t^{d-1} e^{-t} dt$$
$$f_{L_d}(l_d) = \frac{1}{(d-1)!} (-\log l_d)^{d-1}$$

$$\mathrm{Pr}(S_d) = \mathrm{Pr}(L_d \leq x \text{ and } L_{d-1} > x)$$

$$\frac{\mathrm{Pr}(S_d)}{\mathrm{Pr}(L_{d-1} > x)} = \mathrm{Pr}(L_d \leq x \mid L_{d-1} > x)$$

$$= \int_0^1 \mathrm{Pr}(L_d \leq x \mid l_{d-1}) f_{L_{d-1}}(l_{d-1} \mid L_{d-1} > x) \, dl_{d-1}$$

$$= \int_x^1 \frac{x}{l_{d-1}} \frac{f_{L_{d-1}}(l_{d-1})}{\mathrm{Pr}(L_{d-1} > x)} \, dl_{d-1}$$

$$\mathrm{Pr}(S_d) = \frac{x}{(d-2)!} \int_x^1 \frac{1}{l} (-\log l)^{d-2} \, dl$$

$$= \frac{x}{(d-1)!} (-\log x)^{d-1}$$

We now consider the probability of there being any split between 0 and $x$. Let $\pi(d)$ the prior probability of a node being a split, which depends only on the node depth. Then, the probability of any split ocurring between 0 and $x$ is:

$$\mathrm{Pr}(\text{split} \in [0, x]) = \pi(1)(\mathrm{Pr}(S_1) + \pi(2)(\mathrm{Pr}(S_2) + \cdots))$$

$$= \sum_{d=1}^{\infty} \mathrm{Pr}(S_d) \prod_{i=1}^{d} \pi(d)$$

This expression is the probability that the two points are separated, i.e., $1 - k(0, x)$. Unfortunately, from this expression we are able to see that the approximation will quickly diverge away from the Laplace kernel as we increase the preference for low-depth trees, as can be seen in Figure 14. Particularly, for the depth prior both BART and BARK use $\beta = 2.0$ resulting in a vastly different kernel to the Laplace.

## J. Optimizer details

Typical acquisition function optimization is performed using gradient based approaches, such as the second-order method L-BFGS. However, functions sampled from the BARK kernel have zero gradient everywhere, and so these cannot be used. Instead, we formulate the optimization problem as a mixed-integer program, optimizing directly over the tree structure using the formulation in Thebelt et al. (2022a). We use Gurobi 11 to optimize the acquisition function.

This approach comes with additional advantages:

- We can naturally optimize over mixed spaces, as continuous and categorical features are treated equivalently by considering only the tree structure.

- We can include nonlinear constraints.

- In some cases, we can provide a guarantee of global optimality.

### J.1. Optimizer hyperparameters

Table 9 contains list of the hyperparameters that have been set for the solver. We note that MIP solvers have hundreds of parameters, and it would be infeasible to tune them all.

We set the MIP gap to 10%, such that the best acquisition function value is guaranteed to be with in 10% of the global optimum. We found that in practice, the found point is generally much closer to the global optimum, however it takes more time to prove this optimality. We also note that it is not necessary to guarantee global optimality, since gradient-based methods do not guarantee global optimality either.

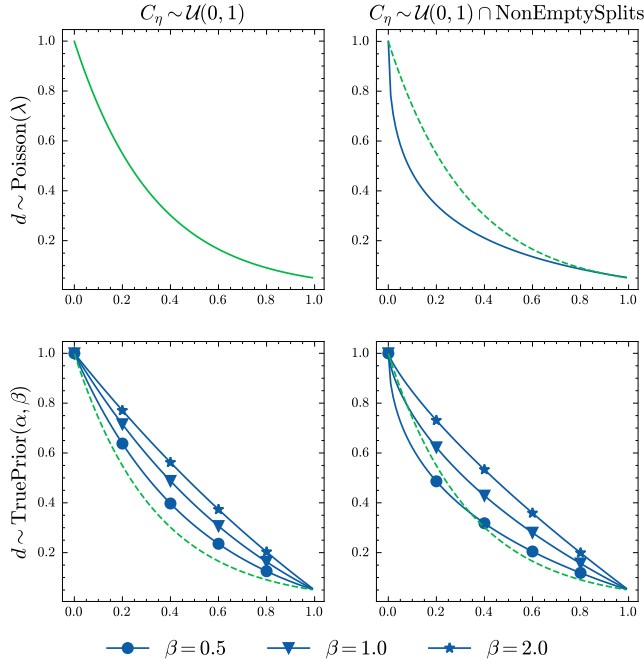

Figure 14: Tree kernel approximation: we compare the effect of strengthening the prior to prefer smaller trees. Indeed, we observe that as $\beta$ increases, the true kernel and the Laplace kernel become very distinct. We set $\lambda = -\ln(1 - \alpha)$. Plots are $k(0, x)$.

| Parameter | Description | Value |
|---|---|---|
| NonConvex | 0 if the problem is convex in the continuous variables; 2 otherwise | 0 |
| MIPGap | Gap between the upper and lower bound of the optimal value | 10% |
| Heuristics | Time spent using heuristics to find feasible points | 20% |
| Time Limit | Maximum time to find the optimum point | 100s |

Table 9: Some settings of the MIP solver parameters.

## J.2. Optimization formulation

See Thebelt et al. (2022a) for further discussion on the formulation of the tree kernel optimization.

We encode the tree model as below. This formulation is based on Thebelt et al. (2022a), with the only difference that the set $\mathcal{T}$ is the set of all trees across all MCMC samples. By using $S$ samples from the tree kernel posterior, we increase the number of constraints in the optimization model by a factor of $S$.

$$\sum_{l \in \mathcal{L}_t} z_{t,l} = 1, \qquad \forall t \in \mathcal{T}, \text{ (9a)}$$

$$\sum_{l \in \mathbf{left}(s)} z_{t,l} \leq \sum_{j \in \mathbf{C}(s)} \nu_{\mathbf{V}(s),j}, \qquad \forall t \in \mathcal{T}, \forall s \in \mathbf{splits}(t), \text{ (9b)}$$

$$\sum_{l \in \mathbf{right}(s)} z_{t,l} \leq 1 - \sum_{j \in \mathbf{C}(s)} \nu_{\mathbf{V}(s),j}, \qquad \forall t \in \mathcal{T}, \forall s \in \mathbf{splits}(t), \text{ (9c)}$$

$$\sum_{j=1}^{K_i} \nu_{i,j} = 1, \qquad \forall i \in \mathcal{C}, \text{ (9d)}$$

$$\nu_{i,j} \leq \nu_{i,j+1}, \qquad \forall i \in \mathcal{N}, \forall j \in [K_i - 1], \text{ (9e)}$$

$$\nu_{i,j} \in \{0,1\}, \qquad \forall i \in [n], \forall j \in [K_i], \text{ (9f)}$$

$$z_{t,l} \geq 0, \qquad \forall t \in \mathcal{T}, \forall l \in \mathcal{L}_t. \text{ (9g)}$$

The objective to be optimized is the UCB acquisition function. The key difference here is that we now have $S$ samples of the mean and standard deviation at each input $\mathbf{x}$, which we sum over to obtain the integrated acquisition function. The equality of the variance is relaxed to an inequality ($\leq$) to provide a second-order cone constraint.

$$\mathbf{x}^*_{\mathrm{lb}}, \mathbf{x}^*_{\mathrm{ub}}, \mathbf{x}^*_{\mathrm{cat}} \in \arg\max_{\mathbf{x}} \sum_{s=1}^{S} \mu^{(s)}(\mathbf{x}) + \kappa \sigma^{(s)}(\mathbf{x}) \tag{10a}$$

$$\mu^{(s)}(\mathbf{x}) = K_{\mathbf{x}\mathbf{X};\theta^{(s)}}(K_{\mathbf{X}\mathbf{X};\theta^{(s)}} + \sigma_y^{2(s)}I)^{-1}\mathbf{y} \tag{10b}$$

$$\sigma^{(s)2}(\mathbf{x}) \leq K_{\mathbf{x}\mathbf{x};\theta^{(s)}} - K_{\mathbf{x}\mathbf{X};\theta^{(s)}}(K_{\mathbf{X}\mathbf{X};\theta^{(s)}} + \sigma_y^{(s)2}I)^{-1}K_{\mathbf{X}\mathbf{x};\theta^{(s)}} \tag{10c}$$

