# OpenReview forum: "BARK: A Fully Bayesian Tree Kernel for Black-box Optimization"
_ICML.cc/2025/Conference — ICML 2025 poster_

### Official Review · Reviewer_GKPb · 2025-03-08

**Overall Recommendation:** 3

**Summary:**

For Bayesian Optimization, the paper proposes to use tree-based functions for the basis for the Gaussian Process prior. The paper outlines the mechanics of this, including:
  * Kernel definition
  * Sampling MCMC method
  * Acquisition definition

The paper then shows that the regression capabilities are reasonable, and experiments with BO over standard synthetic and applied blackbox-optimization benchmarks which mostly consist of mixed search spaces (i.e. containing categorical parameters). The conclusion is that BARK is fairly decent, sometimes optimal compared to previous standard GP baselines (LeafGP, SMAC, Entmoot, etc.)

**Claims And Evidence:**

Generally yes, the paper did in fact satisfy its stated contributions in the intro, i.e.
* Proposing tree-kernel based Bayesian regression method
* Computationally efficient method for training this method (during the log-likelihood maximization phase) and sampling
* Show that it performs reasonably on Bayesian optimization benchmarks.

The only contribution that's missing I feel, is motivating "why propose trees in the first place at all"? Please see "Weaknesses" section for more details. I suspect that BARK would not do as well on a wide variety of continuous-only space benchmarks, and that's fine - but the authors need to make it more explicit where BARK truly shines (mixed spaces).

**Essential References Not Discussed:**

Not that I know of.

**Experimental Designs Or Analyses:**

Yes, paper follows standard Bayesian Optimization evaluation procedures (e.g. running baselines, plotting best observed value over multiple seeded runs) - no red flags here.

Table 1 compares two tree-based regression methods (BART and BARK) but doesn't compare against other standard regression baselines (e.g. Euclidean-GP) - why not? Adding these results establishes much better why tree-based BO works better than euclidean distance-based kernels.

**Methods And Evaluation Criteria:**

Yes, paper uses standard benchmarking functions for BO, well known in literature Synthetic: (TreeFunction, Discrete-Ackley, Rosenbrocks) and Applied (PestControl, CCOBench, etc.). Its regression datasets (UCI) are also well-known.

**Other Comments Or Suggestions:**

For now, I will propose a weak reject - mainly because of the core weakness as I mentioned. But I am fully willing to upgrade my score once the authors respond to this issue.

**Other Strengths And Weaknesses:**

# Weaknesses
While the paper discusses in detail the mechanics of tree-based regression modeling and how it would work in a GP setup, I'm still having a hard time understanding the fundamental conceptual benefits. I.e. what exactly motivated the authors to consider that using trees would be good for Bayesian Optimization (BO)? Is it some combination of:
  * Tree-based functions are better at modeling categorical spaces (which from experience, classic GP-BO methods struggle with)?
  * The structure of the tree-based function (i.e. being very piece-wise and not smooth) helps with modeling non-smooth functions better than using smooth functions as the basis (as a regular Euclidean GP would)?

If this is the case, I would strongly urge the authors to motivate these more, instead of jumping into the mechanics of the BARK method. Perhaps also provide figures on how this new BARK-GP regresses on a 1D categorical / discrete function better than a RBF-GP?

EDIT: My weaknesses have been resolved. The authors have made their contributions over mixed spaces much more explicit.

**Questions For Authors:**

Please see my overall weaknesses section.

**Relation To Broader Scientific Literature:**

Mixed search spaces are in general have been fairly difficult to model with general euclidean-based GP kernels. This is because of having to deal with continuity issues / it's difficult to capture the shape of the objective very well with a smooth model designed originally for continuous spaces.

So this paper does make a solid contribution to this area of Bayesian Optimization.

**Theoretical Claims:**

Did not check carefully Section F (theoretical regret bounds).

---

> ### Author Rebuttal · Authors · 2025-03-31
>
> Thank you for your careful review, and for recognizing our contribution to mixed-space Bayesian optimization.
>
> > what exactly motivated the authors to consider that using trees would be good for BO
>
> We agree with the reviewer that we should better motivate BARK to ensure practitioners use BARK appropriately. Our revised paper will include the requested figures and the following discussion.
>
> Our greatest motivation for using trees is for modeling - and specifically optimizing over - mixed feature spaces. Purely continuous spaces are well-modeled by standard Euclidean GP kernels. Purely categorical spaces can be addressed with multi-armed bandit approaches. However, many real-world problems have mixed domains. Trees have strong modeling performance in mixed spaces. Moreover, we can define a MIP to optimize the AF exactly and we can formulate this MIP so that it is effectively solved with off-the-shelf software. The resulting BO procedure does not require approximations during the optimization step.
>
> Modeling with trees offers several additional benefits. First, we are able to model non-stationary functions, where the lengthscale changes throughout the domain. We demonstrate this behavior in [Rebuttal Figure 1](https://anonymous.4open.science/r/bark-rebuttal-8830/Fig\_1D\_toy\_continuous.pdf). This is especially desirable in high dimensions, where a locally short lengthscale will not harm BARK's global uncertainty quantification.
>
> Second, we assume prior correlation between values in categorical features due to the nature of splitting rules in trees.
> The indicator kernel typically used for categorical features with GPs (Ru et al., 2020) assumes that each value is independent. BARK is able to capture correlations between values, as demonstrated in [Rebuttal Figure 3](https://anonymous.4open.science/r/bark-rebuttal-8830/Fig\_1D\_toy\_categorical.pdf). Trees therefore have a more expressive prior on functions over categorical values.
>
> Some black-box functions are indeed better modeled by trees, as demonstrated in the MAX benchmark where a grid-based AF optimization is used for all methods. However, other benchmarks are better modeled with the assumptions of smoothness provided by Euclidean GP kernels.
>
> > Table 1... doesn't compare against other standard regression baselines
>
> Section 7.1 aims to show that BART and BARK have similar modeling abilities, and that our kernel perspective on BART still leads to strong regression performance. Since our only point is that BART and BARK are similar with respect to regression, we did not compare to a wide array of regression models. However, we agree that including RBF-GP would provide more context and the revised paper will include an extended version of the table: [Rebuttal Table 2](https://anonymous.4open.science/r/bark-rebuttal-8830/Tab\_Regression\_with\_GP\_RBF.pdf). Additionally, as noted in our reply to Reviewer zyg8, we will signpost the purpose of Table 1 more clearly.
>
> > I suspect that the tree-based kernel does indeed perform well on mixed search spaces, but the authors need to make this motivation more explicit. This can be boosted better with... benchmarking on continuous-only functions
>
> Thanks for helping us make our motivation more explicit.
> Indeed, we do not expect BARK to outperform an RBF kernel in continuous-only BO. As explained by the reviewer, these continuous-only BO problems may be smooth, and therefore have a higher prior probability under an RBF kernel. The reviewer is correct that our original submission focuses on mixed spaces where we expect BARK to be more suitable. We also agree that adding continuous-only benchmarks offers a more complete view. We will provide two continuous-only benchmarks in the revised paper: Hartmann (6D) and Styblinski-Tang (10D), which we show in [Rebuttal Figure 4](https://anonymous.4open.science/r/bark-rebuttal-8830/Fig\_continuous\_benchmarks.pdf). Despite BARK's non-smooth assumption, we still observe reasonable performance from BARK in this setting.
>
> > how this new BARK-GP regresses on a 1D categorical / discrete function
>
> To motivate using trees, our revised paper will include an example of regression on both a 1D discrete function and a 1D categorical function: [Rebuttal Figure 2](https://anonymous.4open.science/r/bark-rebuttal-8830/Fig\_1D\_toy\_discrete.pdf) and [Rebuttal Figure 3](https://anonymous.4open.science/r/bark-rebuttal-8830/Fig\_1D\_toy\_categorical.pdf). We will explain in the revised paper that BARK may not necessarily provide better regression for discrete, ordinal features (which are simply continuous features sampled on a grid) compared to Euclidean GPs, but that BARK provides a method of optimizing over such features without approximations.

---

> > ### Comment · Reviewer_GKPb · 2025-04-01
> >
> > Thank you for the response, and my weaknesses have been resolved. The authors have made their contributions over mixed spaces much more explicit. Thus I upgrade my score.

---

### Official Review · Reviewer_PzbN · 2025-03-14

**Overall Recommendation:** 3

**Summary:**

The paper introduces a new tree-based surrogate model for use in Bayesian optimization. It extends of prior work in tree-based regression, particularly BART, to be more suitable for acquisition function optimization and Bayesian optimization. The tree model is fully Bayesian, including MCMC over tree structures. The acquisition function is an integer program. The method performs well on real mixed-variable problems.

## update after rebuttal
No change

**Claims And Evidence:**

The claims of the paper were well supported.

**Essential References Not Discussed:**

N/A

**Experimental Designs Or Analyses:**

The empirical evaluation was typical for a BO paper like this.

**Methods And Evaluation Criteria:**

The method was well motivated and very clearly described. There was a strong set of ablations and results for understanding model performance. The choice of benchmark problems was reasonable.

**Other Comments Or Suggestions:**

Fig. 5 shows error bars but doesn't say what they are.

The paper states that software the model will be released. It seems that this software will depend on Gurobi, which is very expensive for many people. If the authors would like to maximize the impact of their work, it would be worth the effort to introduce an interface for using a free/open source solver.

**Other Strengths And Weaknesses:**

I found the ideas in the paper to be an interesting, novel, and likely useful. The one major weakness in the paper is the complete lack of discussion of wall-time for BO. I expect this to be slow due to the combination of MCMC and integer programming (both of which tend to be slow). But I do not know if this is unusably slow or not, as the only wall-times given are for sampling, just for the regression problems and compared only to BART (Table 5 in the supplement). I expect to see total wall time for the full BO loop, compared with wall time for the other BO methods; and for that total wall time to be broken down into MCMC time vs. Gurobi time. I cannot recommend the paper very enthusiastically without that result as based on what I see now, the method may not be practical. It seems very promising though.

**Questions For Authors:**

How does total BO wall time compare to all of the other baselines methods, and how much of that wall time is spent doing MCMC and how much is spent in Gurobi?

**Relation To Broader Scientific Literature:**

The framing with the broader scientific literature was well written and appropriate.

**Theoretical Claims:**

No

---

> ### Author Rebuttal · Authors · 2025-03-31
>
> We kindly thank the reviewer for their thoughtful review, and for recognizing the potential of our method.
>
> > lack of discussion of wall-time for BO
>
> As mentioned in Appendix G.1, we limit each MIP optimization to 100 seconds. However, we agree that a more complete comparison is highly relevant. Please see requested results in [Rebuttal Figure 5](https://anonymous.4open.science/r/bark-rebuttal-8830/Fig\_optimization\_time.pdf) for a selection of 3 benchmarks. We will also include this figure and the following discussion in the main paper:
>
> BARK is slower than competing methods, taking ~50s to fit the model, and 100s to optimize. This is due to the combination of the expensive MCMC procedure, and the large optimization formulation. This is comparable to the fitting time for BART, and the time taken to evaluate BART on a grid of 2^14 points. BARK is best applied in BO settings where the objective is expensive to evaluate (e.g. taking at least several minutes, or having a large associated financial cost). Note that the PestControl and MAX (material design) benchmarks reflect examples of such black-box functions. For settings where experiments are cheap and/or function evaluations are quick, we recommend alternate methods.
>
> > Fig. 5 shows error bars but doesn't say what they are.
>
> Thanks for noting our omission: the revised paper will state explicitly in the figure caption that the bars are the 25th and 75th percentile of regret achieved across the 20 runs.
>
> > this software will depend on Gurobi, which is very expensive for many people
>
> Gurobi does provide a free academic license, but we recognize that requiring a licensed software will limit the reach of this work. Gurobi is a very strong optimizer for MIP problems, and open source solvers would take longer to maximize the AF. However, we would still be interested in providing such an interface, especially if there is community interest.

---

> > ### Comment · Reviewer_PzbN · 2025-04-04
> >
> > Thanks, 150s is pretty reasonable for a lot of tasks and makes this a useful method.

---

### Official Review · Reviewer_zyg8 · 2025-03-18

**Overall Recommendation:** 4

**Summary:**

The paper proposes a combination of forest kernel GPs and Bayesian tree models (BART) specifically tailored for the use in Bayesian Optimisation (BO). The main idea is to directly optimise the expected acquisition function values over the posterior distribution of the kernel parameters. This is done through posterior samples of those parameters obtained via a Metropolis Hastings Markov Chain Monte Carlo (MCMC) algorithm. The final optimisation of the mean acquisition function can apparently be done efficiently via a mixed integer programming (MIP) approach. Compared to a BART-BO approach where posterior sampling is performed to obtain individual trees, the proposed approach is said to lead to a much improved sample efficiency per MCMC sample.

**Claims And Evidence:**

The main claim is that the proposed method tends to outperforms similar method in their blackbox optimisation performance. The claim makes sense conceptually and experimental evidence is provided.

**Essential References Not Discussed:**

None to my knowledge.

**Experimental Designs Or Analyses:**

Overall the experiments cover a reasonable range of test cases and the results look sensible. Specifically for the comparison to BART I am wondering though whether the comparison does not have to be refined. This is because we principally allow for large computation times of BO methods. Hence, the key claim of an improved efficiency seems to require some quantification in terms of wall clock computation time. Having less sample efficiency alone is not necessarily a big disadvantage if one can allow a lot of samples and the complexity per sample can differ.

**Methods And Evaluation Criteria:**

The optimisation benchmark problems are standard in the field. The choice of UCI datasets, which are not used for optimisation, to investigate the model fit is questionable though, as the main objective of the paper is to explain differences in optimisation performance. Here the paper would benefit from a more stringent formulation of objectives (hypotheses to be tested) of the experimental evaluation.

**Other Comments Or Suggestions:**

See other strength and weaknesses. It would really be useful to see more compactly the various parameters, especially the kernel parameters that are ultimately sampled.

**Other Strengths And Weaknesses:**

The paper makes a laudable approach to compactly yet accurately present the extensive amount of ideas, known and novel, that are required to understand the subject matter. Yet to my perception it is still hard to follow and to comprehensively see all the difference to alternative approaches. Could it be useful to have a hierarchical presentation of a general form of the model used in this and prior works, and from there develop their differences and commonalities?

The MIP approach for acquisition function optimisation seems to be very interesting in its own right. Unfortunately, there seems to be no room to discuss this in the main paper.

**Questions For Authors:**

None.

**Relation To Broader Scientific Literature:**

The paper makes an excellent job in surveying the key literature in an integrative and insightful way.

**Theoretical Claims:**

There is an interesting theoretical claim about the convexity of the expected acquisition function value at query points in terms of predictive mean and variance. This and other claims might be proven in the appendix, which I did not have time to check.

---

> ### Author Rebuttal · Authors · 2025-03-31
>
> We thank the reviewer for their thorough review, and for identifying the strengths of our work.
>
> > The choice of UCI datasets
>
> Section 7.1 demonstrates that the BARK model performs similarly in regression to the BART model. BART is typically used in regression settings with tabular datasets, so the purpose using these UCI benchmarks is to be fair to BART. This section demonstrates that the improved performance of BARK in BO is due to the acquisition function optimization, and not any superior modeling capability.
>
> Agreed with the reviewer that we should state the purpose of  Section 7.1 more clearly: the revised paper will clearly signpost our objective of comparing regression tasks where BART is already known to be strong.
>
> > the convexity of the expected acquisition function
>
> Thebelt et al. (NeurIPS, 2022) show the convexity of the nonlinear part of the acquisition function: see their development in their Section 4. The acquisition function (AF) itself is not convex in the input space, but only with respect to the mean and variance, which leads to improved performance with mixed-integer solvers.
>
> > for the comparison to BART I am wondering though whether the comparison does not have to be refined... the key claim of an improved efficiency seems to require some quantification
>
> Many BART samples are required to compute the predictive variance, e.g. experiments in Chipman et al. (2010) use at least 1000 function samples. We agree with the reviewer that, in regression, this lower sample efficiency may not be a disadvantage. However, the high number of samples means that it is infeasible to formulate an optimization problem over the BART AF, as the number of variables and constraints required to encode these tens of thousands of trees would be too great. The large number of samples therefore necessitates a grid-based approach to optimizing the AF. Our revised paper will clarify the relevance of the 'sample efficiency' claim in Section 5.2.
>
> Furthermore, the required density of the grid on which BART is evaluated increases exponentially with the problem dimension, and so it is not feasible to allow for a sufficiently dense grid in high dimensional problems. The grid sized used in our experiments was chosen to match the wall clock time of the BARK optimization (see [Rebuttal Figure 5](https://anonymous.4open.science/r/bark-rebuttal-8830/Fig\_optimization\_time.pdf), where evaluating BART at 2^14 gridpoints takes the same wall-clock time as the BARK optimization).
>
> > Could it be useful to have a hierarchical presentation of a general form of the model used in this and prior works
>
> Thank you for the suggestion - we will add to the paper a summary of the existing literature (see [Rebuttal Table 3](https://anonymous.4open.science/r/bark-rebuttal-8830/Tab\_literature\_comparison.pdf)),  that highlights key similarities and differences between the various BO methods covered in the literature review.
>
> > no room to discuss [the MIP approach] in the main paper
>
> Thebelt et al. (NeurIPS, 2022) develop a detailed discussion of the MIP. We have extended the optimization model, and will clarify these improvements in Appendix G. The contributions we will clarify in the revised version include: extending the MIP to include multiple tree-structure samples and linking the kernel samples in the (approximate) integrated AF.
>
> > see more compactly the various parameters
>
> Thanks for the nice idea, which helps us improve clarity. We will add this summary of the BARK parameters (see [Rebuttal Table 1](https://anonymous.4open.science/r/bark-rebuttal-8830/Tab\_BARK\_parameters.pdf)) to the paper.

---

### Decision · Program_Chairs · 2025-05-01

**Decision:**

Accept (poster)

**Comment:**

The authors propose a Bayesian tree model estimated via MC that can be used as a kernel for a GP.  The AF is optimized via Gurobi.  They demonstrate that this kernel can yield improvements over a few baselines.  All reviewers provided at least an accept rating post-rebuttal. It was an interesting and gratifying paper to read.

Two major weaknesses were pointed out by the reviewers: one was a question (PzbN) about the walltime (which had been evaluated and shared during the rebuttal; wall time is an order of magnitude slower (even w/ a limited AF optimization budget and the model itself would be prohibitively slow in high dimensions).  The other was a comment re: details of the AF optimization (GKPb), which while not flagged by any reviewers, is likely big issue in the empirical evaluation.

Here is my concern with the paper: In short, GP-RBF likely has much stronger empirical performance than reported if the authors were to use a more appropriate method for optimizing the AF.  GP-RBF has the best performance when the search space is enumerable (cf, XGBoost experiment).  When the cardinality becomes too large or the space is mixed, the authors use continuous relaxation for the discrete variables, which is well-documented to not work well.  A simple approach is to instead do alternating local search (see e.g., [2,4,5]), or alternatively gradient-based optimization via probabilistic reparameterization [3].  I expect this approach to be much more effective in the settings studied as well as higher-dimensional problems (studied in some of the papers below) which—as far as I understand—would be impractical for BARK.  It's also possible that instead of 1-hot encoding the categorical, the use of categorical kernels [1,2] or embeddings [6], could improve performance.

I expect that in the CR, the authors will utilize the alternating local search  instead of using the continuous relaxations and rounding.

The easiest way to accomplish this might be to just swap out BoFire for Ax, which should use the same priors and does alternating local search for mixed search spaces (via BoTorch's `optimize_acqf_mixed_alternating()`).  It wasn't clear from the text how the authors were modeling and optimizing the AF for ordered categorical or integer-valued inputs.  You may wish to double check that this information is being passed into BoFire or Ax as such so that they can be encoded directly into the RBF, rather than one-hot encoding.

As per the author-reviewer discussion, I also expect more wall-time discussion and experiments, including plots or tables that include overall walltimes for BARK and baselines.  Breaking these timings out by fitting and optimization time would be insightful, given reviewer feedback. It would be nice to see more benchmarks problems to illustrate scaling wrt the input dimensionality.  Some of the papers below could provide inspiration for such problems:

- [1] Ru et al. ICML 2020. propose the use of a categorical kernel (IIRC used in the two papers below)
- [2] Wan et al. ICML 2021 use alternating local search, categorical kernels
- [3] Daulton et al. NeurIPS 2022 optimize over ensembles of exact designs via gradient-based optimization
- [4] Papenmeiher et al. NeurIPS 2023. demonstrate pathologies of common mixed-variable benchmark problems, and propose an improved embedding model for the discrete spaces
- [5] Zhang et al.  Technometrics 2019. propose the LVGP, which uses a 1 or 2D embedding of categorical variables; this is fairly easy to implement in GPyTorch by adding an embedding layer.

Finally, none of the reviewers (or the meta-reviewer) verified the theoretical results in detail, so I encourage the authors to try to obtain some additional feedback on those before wrapping up the CR.